# Spatial transition tensor of single cells

Peijie Zhou [1,4,5,6], Federico Bocci[1], Tiejun Li[2] & Qing Nie [1,3] ✉

Spatial transcriptomics and messenger RNA splicing encode extensive spatiotemporal information for cell states and transitions. The current lineage-inference methods either lack spatial dynamics for state transition or cannot capture different dynamics associated with multiple cell states and transition paths. Here we present spatial transition tensor (STT), a method that uses messenger RNA splicing and spatial transcriptomes through a multiscale dynamical model to characterize multistability in space. By learning a four-dimensional transition tensor and spatial-constrained random walk, STT reconstructs cell-state-specific dynamics and spatial state transitions via both short-time local tensor streamlines between cells and long-time transition paths among attractors. Benchmarking and applications of STT on several transcriptome datasets via multiple technologies on epithelial–mesenchymal transitions, blood development, spatially resolved mouse brain and chicken heart development, indicate STT's capability in recovering cell-state-specific dynamics and their associated genes not seen using existing methods. Overall, STT provides a consistent multiscale description of single-cell transcriptome data across multiple spatiotemporal scales.

The advances of single-cell gene expression profile techniques have provided an unprecedented resolution to dissect cell-fate decisions. Metrics such as similarity or distance on a low-dimensional manifold are applied to single-cell RNA sequencing (scRNA-seq) data to infer dynamic properties such as pseudotime ordering[1,2], network abstraction[3] or cellular random walk analysis[4,5]. Leveraging both unspliced and spliced counts, the RNA velocity methods[6,7] explicitly model the dynamics of messenger RNA (mRNA), projecting the future spliced states of cells onto scRNA-seq data to reveal the directionality of cell-fate determination[8], and also to improve trajectory inference[9–11], low-dimensional embedding[12,13] and gene regulatory network inference[14,15].

Spatial transcriptomics measures additional spatial information at individual cells or spots of a small group of cells, allowing analysis of heterogenous cell states in space[16,17]. To infer temporal dynamics within spatial transcriptomics, SpaceFlow[18] uses proximity information to constrain the cell embedding and pseudotime ordering for spatial consistency. SIRV[19] develops a spatially resolved RNA velocity approach, by improving estimation of unspliced and spliced mRNA using reference scRNA-seq counterparts to enrich the spatial transcriptomics gene expression matrices.

While RNA velocity has been widely used, fundamental challenges remain for reconstructing robust spatiotemporal dynamics[20]. For example, multilineages or multiple meta-stable states[21–23] in complex spatial tissues cannot be captured by the current models, as spliced and unspliced transcript levels may diverge due to nonlinear gene regulation or multicellular signaling. In addition, the time scale of mRNA splicing is within minutes or hours[24,25], during which the current RNA velocity model converges to one global equilibrium, however, cell-state transitions may span from days to weeks, (for example, in hematopoiesis[8,20,25]). While cell-specific gene expression rates may be used to accommodate a continuous cell-fate commitment process[25,26], additional measurements, such as metabolic labeling[27–29], are needed[25] and difficult to obtain, for example, in spatial transcriptomics. Last, the current major RNA velocity methods are only focused on the velocity

[1]Department of Mathematics, University of California, Irvine, Irvine, CA, USA. [2]LMAM and School of Mathematical Sciences, Peking University, Beijing, China. [3]Department of Cell and Developmental Biology, University of California, Irvine, Irvine, CA, USA. [4]Present address: Center for Machine Learning Research, Peking University, Beijing, China. [5]Present address: AI for Science Institute, Beijing, China. [6]Present address: National Engineering Laboratory for Big Data Analysis and Applications, Beijing, China. ✉e-mail: qnie@uci.edu

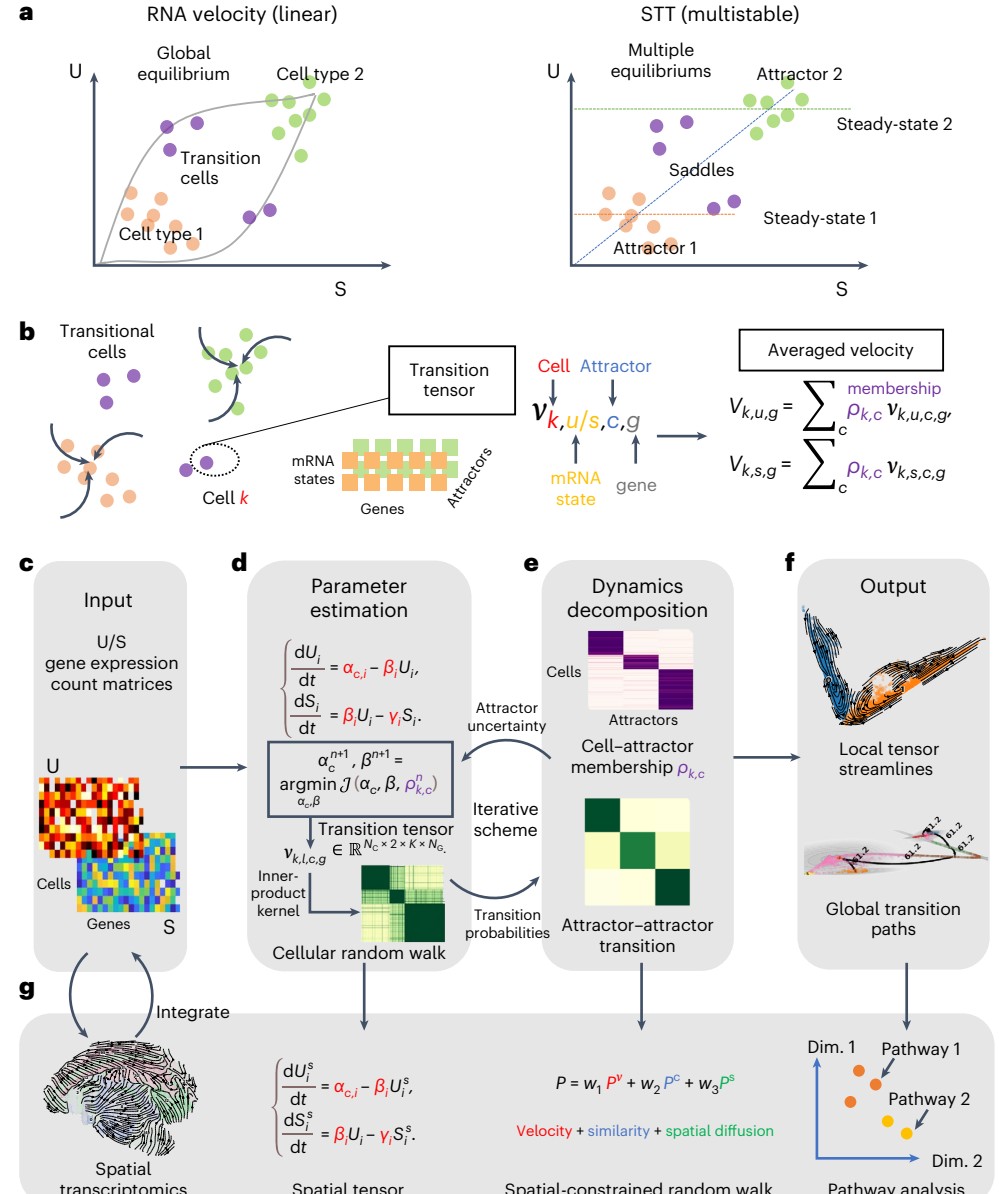

**Fig. 1 | Overview of STT. a**, Comparison between the RNA velocity (linear and single equilibrium) versus STT tensor model (multistable and multiple attractors). **b**, Definition of transition tensor and induced RNA velocity by averaging cell's membership in different attractors. **c–f**, Workflow of the STT. **c**, The input U and S count matrices. **d,e**, Iterative scheme between kinetic parameter estimation of transition tensor (**d**) and dynamics decomposition and coarse-graining (**e**). **f**, Output of STT. **g**, Analysis of spatial transcriptomics data using STT where the spatial-similarity kernel based on spatial cell coordinates is combined with the tensor-induced and gene expression-induced kernel to infer a cell's membership in attractors. In pathway similarity graph, Dim. denotes the coordinates in reduced dimensions.

of spliced counts, omitting the velocity of unspliced counts that are closely linked to gene regulation[15], which could provide further information about 'attraction force' into certain cell state.

The multiscale cell attractor theory[30–35] provides a natural tool to model dynamics across different time scales and resolutions, as well as account for the multistable states. In such a theory, the temporal change of gene expression and their mutual regulations are modeled as dynamical system composed of a set of differential equations. The stable cell types correspond to multiple locally stable fixed point of dynamical system under mild perturbation of gene regulation (that is, multistable states) where the cell states of expression are 'trapped', and the highly plastic transitional cells are modeled as 'saddle point' of the system, such that the cell could make state transitions through certain direction. Using such an approach, MuTrans[5] coarse-grains scRNA-seq

data at different scales to identify attractors and saddle points, allowing description of short-time fluctuations of cells around attractors locally while capturing long-time scale transitions of cells among multiple attractors with saddle points in between. The Gaussian-like kernel in MuTrans confines its scope to equilibrium and ergodic systems[4,5]. For nonequilibrium systems, using RNA velocity as input, CellRank[8] constructs a cellular random walk using a velocity kernel followed by coarse-graining analysis and Dynamo[25] fits the discrete RNA velocities using continuous functions for attractor geometry and transition analysis. However, in these methods, the linear RNA velocity model is incompatible with the presence of multistable attractors inherited in the data, leading to inconsistency between the transition velocity and downstream analysis. In addition, such approaches cannot be used directly for spatial transcriptome data.

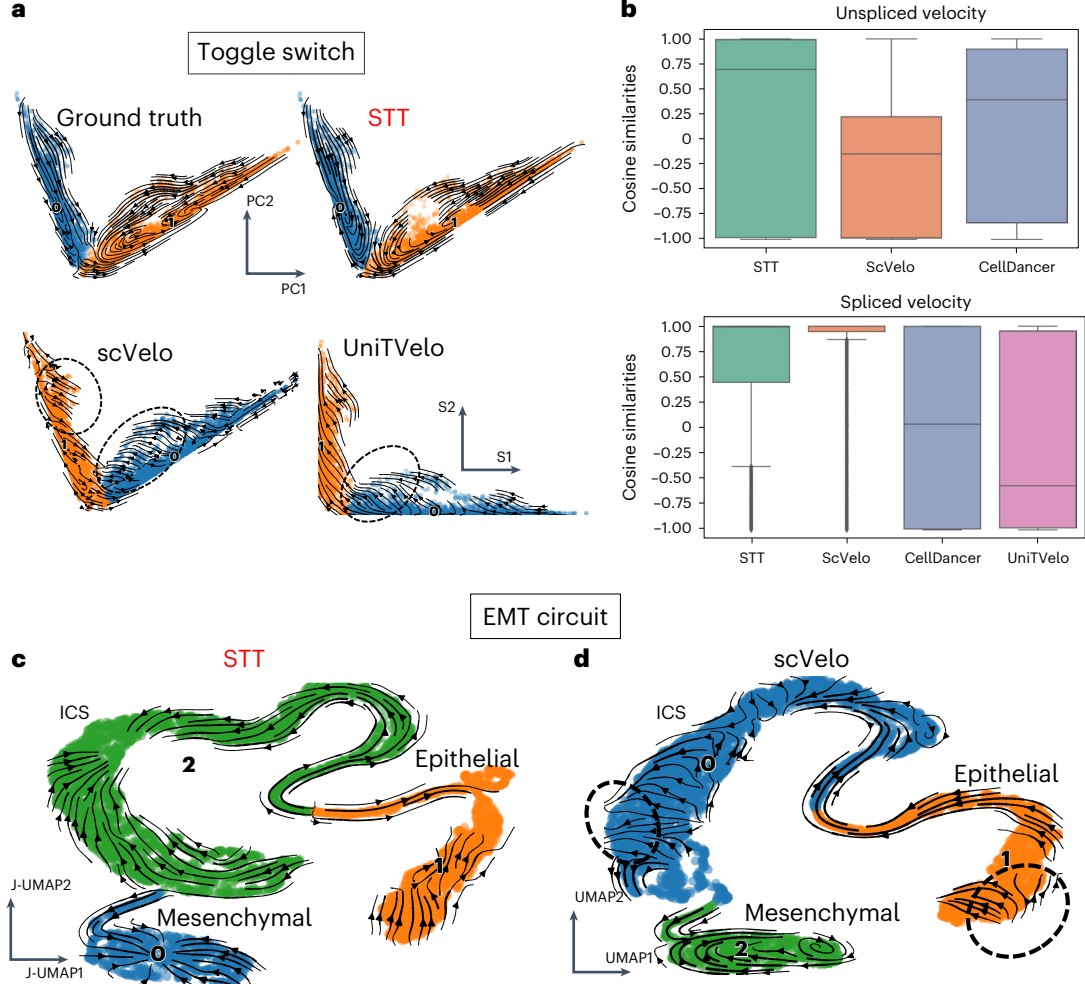

**Fig. 2 | Benchmarking of STT in simulation datasets of toggle-switch and EMT circuits. a**, Comparison between streamlines of STT and other methods for toggle-switch dataset. The cells are colored by attractor in STT, or Leiden clustering results in scVelo and UniTVelo. The STT, scVelo and ground-truth results are embedded in PCA on joint spliced and unspliced counts, and UniTVelo result is plotted on the coordinates of spliced counts. **b**, The box plots across all cells ($n$ = 10,010) of cosine similarity between calculated velocity and ground truth in different methods. The central box represents the interquartile range, from the 25th percentiles (bottom bounds) to 75th percentiles (top bounds), and horizontal line within the box indicates the median (50th percentile).

The whiskers stretch out to the values that fall within 1.5 times the interquartile range from the lower and upper quartiles. The dots indicate outliers. **c,d**, Comparison between streamlines of STT and other methods for synthetic EMT circuit dataset. **c**, The cells are colored with attractor assignment by STT, and the low-dimensional embedding is the UMAP based on the joint of spliced and unspliced counts. The streamlines are visualized using the averaged velocity over attractors. **d**, The cells are colored with Leiden clustering output, and the low-dimensional embedding is the UMAP of spliced counts only. The streamlines are visualized using RNA velocity.

Here we present a spatial transition tensor (STT) approach to reconstruct cell attractors in spatial transcriptome data using unspliced and spliced mRNA counts, to allow quantification of transition paths between spatial attractors as well as analysis of individual transitional cells. Unlike the linear RNA velocity model with one global equilibrium (Fig. 1a), STT assumes the coexistence of multiple attractors in the joint unspliced (U)–spliced (S) counts space, with cells making transitions between attractor basins (Fig. 1a,b). A four-dimensional transition tensor across cells, genes, splicing states and attractors is constructed, with attractor-specific quantities associated with each attractor basin (Fig. 1b). By iteratively refining the tensor estimation and decomposing the tensor-induced and spatial-constrained cellular random walk (Fig. 1c–e,g), STT connects the scales between local gene expression and splicing dynamics as well as the global state transitions among attractors. Furthermore, STT ranks genes that are mostly relevant to the multistable expression patterns, and categorizes pathways with similar STT properties (Fig. 1g). By studying both nonspatial and spatial datasets, we demonstrate STT's unique capability to uncover multistable attractors of cells and transition properties occurring at different spatiotemporal scales.

## Results

### Overview of STT

The inputs to STT are the single-cell gene expression matrices of both S and U counts (Fig. 1c), and the cell annotations (or membership) that serve as initial guess on what cell state they belong to. In addition, the spatial coordinates of each cell (or spot) are also required for spatial transcriptomic data. Through an iteration between parameter estimation and dynamics decomposition, STT constructs an attractor-wise velocity tensor named transition tensor of shape $\mathbb{R}^{N_C \times 2 \times K \times N_G}$, where $N_C$ denotes the number of cells, $N_G$ the number of genes and $K$ the number of attractors. Other quantities of tensor-based dynamics, including the memberships of cells in the attractors, transition probabilities and transition paths, are subsequently obtained in this construction (Methods).

STT uses the following stochastic model of gene expression and splicing dynamics

$$\begin{cases} dU_i = (f_i(t, S_1, ..., S_{N_G}) - \beta_i U_i)dt + \sigma_i dW_{i,t}, \\ dS_i = (\beta_i U_i - \gamma_i S_i)dt + \sigma_i dZ_{i,t}, \end{cases} \quad (1)$$

where $U_i$ and $S_i$ are the unspliced and spiced counts for gene $i$. The nonlinear function $f_i(t, S_1, ..., S_{N_G})$ models how other genes regulate the production rate of gene $i$. The system can possess multiple fixed points or attractors representing the different cell states. The parameter $\beta_i$ represents the mRNA splicing rate and $\gamma_i$ is the spliced mRNA degradation rate. The independent Wiener process terms $W_{i,t}$ and $Z_{i,t}$ represent the noise in gene expression. Such stochasticity may induce the noise-induced cell-state transitions among multistable attractors at a longer time scale than splicing dynamics.

When most cells are located within the multiple attractor basins that correspond to the different cell states, with a small fraction of cells making transitions across the saddle points[5] (a natural assumption on the cell distribution), the unspliced mRNA production term can be expanded and approximated to its linear expansion, thus introducing the attractor-dependent mRNA transcription rate (Fig. 1d and Methods). Such expansion allows robust estimate of the parameters, and initializes assignment of the attractor-wise velocities for each cell, which we call transition tensors (Fig. 1d and Methods).

By constructing an inner-product velocity kernel (Fig. 1d, Methods and Supplementary Note 1), the tensors provide a cellular random walk description that is asymptotically consistent with continuous stochastic differential equation (SDE) (that is, equation (1)). Combining with the Gaussian kernel of gene expression similarity and cell spatial coordinates (Fig. 1g and Methods), the constructed cellular random walk equips cells in each attractor with consistent velocity, transition direction and similar gene expression. In addition, the constructed random walk encourages cells to be more likely to make transitions to other spatially adjacent cells in the physical space. Through coarse-graining and decomposing the random walk on attractor levels, the cells' membership functions for different attractors are then obtained (Fig. 1e and Methods). In each iteration between the tensor model construction and the random walk decomposition, the updated membership function improves the parameter estimation in equation (1) by incorporating attractor uncertainty (Methods). The genes, whose dynamics are most consistent with the attractor property in the $U$–$S$ space, are then identified during iteration (Methods). A monitor module is included, with regularization and early stopping strategies that can improve the robustness of iteration through the user's control (Methods). Finally, the tensor streamlines to describe the attractor details, as well as the coarse-grained transition paths to depict long-time transitions, are projected on a low-dimensional dynamical manifold to show the cell-state transitions (Fig. 1f and Methods).

### Benchmarking STT in recovering multistable cell states

We first applied STT to analyze two synthetic datasets based on simulating multistable systems. In the bistable toggle-switch circuit, the streamlines of averaged velocities over attractors in STT demonstrate clearer structures of the two attractors than the streamlines of RNA velocity and other methods (Fig. 2a and Supplementary Fig. 1). While RNA velocity streamlines computed by scVelo[7] and UniTVelo[36] tend to diverge from the attractor locations, STT streamlines converge toward the attractors, thus providing a more interpretable representation of the toggle-switch landscape (Fig. 2a and Supplementary Fig. 1). Moreover, STT computes an entropy value to distinguish between stable cells near fixed point and transitional cells across saddle points (Supplementary Fig. 1). As shown in both components of transition tensors with streamlines (Supplementary Fig. 1), only when the unspliced and spliced quantities are considered together can both attractor basins be

revealed. Although the spliced tensors are consistent with the standard RNA velocity (Fig. 2a), which depicts transitions between the attractors, the unspliced tensors naturally introduce an 'attraction force' that 'pulls' cells toward the center of each attractor, as compared to the streamlines of cellDancer[37] where the cells are attracted to the 'ends' within attractor (Supplementary Fig. 1). The unspliced counts provide a measurement on the level of 'attraction' in STT for an attractor of cell state. To further benchmark the accuracy of STT, we compared the cosine similarity between STT unspliced or spliced tensor components and the ground-truth velocities from the model, and found that STT ranked top in estimating both spliced and unspliced velocities (Fig. 2b). In addition, the performance of STT shows a good level of robustness when subsampling the dataset (Supplementary Fig. 1).

Next, we analyzed the simulated gene regulation circuits during epithelial–mesenchymal transition (EMT), where three attractors, denoted as epithelial (E), mesenchymal (M) and intermediate cell state (ICS), may coexist, in some parameter ranges (Methods). Compared to the RNA velocity calculated by scVelo (Fig. 2), the STT average velocities (Fig. 2c) clearly recover these three attractors. Overall, STT is able to reconstruct the complex multistable details in single-cell gene expression datasets.

### STT highlights ICSs in fate decision

We next analyzed the scRNA-seq data in the EMT induction experiment of human lung A549 cell lines, including a temporal series of snapshots collected from the first 7 days after TGFB1 treatment[38]. STT identifies three attractors, namely E, ICS and M, consistent with the order of timepoints in data collection (Fig. 3a,b and Supplementary Fig. 2). Moreover, cells nearby the ICS attractor, mainly collected at 8 h or 1 day after induction (Fig. 3b), have higher entropy values (Fig. 3c), thus indicating that this state is more plastic than epithelial (day 0 and 8 h) and mesenchymal (after day 3) states. This is in good agreement with the proposed phenotypic plasticity of intermediate epithelial and/or mesenchymal states in cancer[39].

Using the transition vector to predict the transition paths connecting attractors in the epithelial–mesenchymal landscape, we find that the transition probability flux from E to M always goes through the ICS (Fig. 3a). In other words, epithelial cells undergoing EMT never directly switch to a mesenchymal state, but rather acquire intermediate traits first. The unspliced and spliced counts often exhibit multistability of the attractors (Fig. 3e and Supplementary Figs. 2 and 3). The genes with high multistability scores possess varying expression levels in both unspliced and spliced counts within various attractors, and show a gradual change during E–ICS–M transitions. While the highly ranked multistable genes such as ITGA11, are not significantly detected by differential gene expression analysis as top-scored marker genes for attractors (Supplementary Fig. 3), they are found important in promoting EMT transitions and tumor progression[40]. While the tensor streamlines of splicing dynamics demonstrate the overall direction from E to M via ICS, which is also consistent with the UniTVelo results (Supplementary Fig. 2), the gene expression dynamics of unspliced counts as well as in the joint $U$–$S$ space predicted by STT as well as cellDancer both suggest that cells are 'attracted' to the ICS basins during EMT (Fig. 3f and Supplementary Fig. 2). This is also consistent with the CellRank absorption probability analysis based on tensor-induced multistability kernel (Fig. 3d). Together, the tensor components along with the global transition paths analysis highlight the ICS as a distinct attractor basin, serving as the hub state during EMT.

**Fig. 3 | Multistability of EMT in A549 cell lines with TGFB1 induction. a**, The global transition path analysis of EMT. Cells are embedded in the constructed transition coordinates (trans. coord.) of dynamical manifold and the number indicates fraction of transition flux. Cells are colored by STT attractor. **b**, Transition coordinates with cells colored by collection time. **c**, Violin plot of cell-membership entropy in different attractors. **d**, Absorption probabilities of cells into different attractors using multistability kernel induced random walk by STT. **e**, Top genes that are consistent with the multistability of attractors in EMT. **f**, The streamlines of various components of transition tensors, including the attractor-averaged and attractor-specific tensors. The low-dimensional embedding is the UMAP of both spliced and unspliced counts. In the left panel, the cells are colored by the attractor assignment. In the right panel, the cells are colored by their membership in each attractor, and only the tensors of cells whose memberships are greater than 0.2 in the attractors are shown.

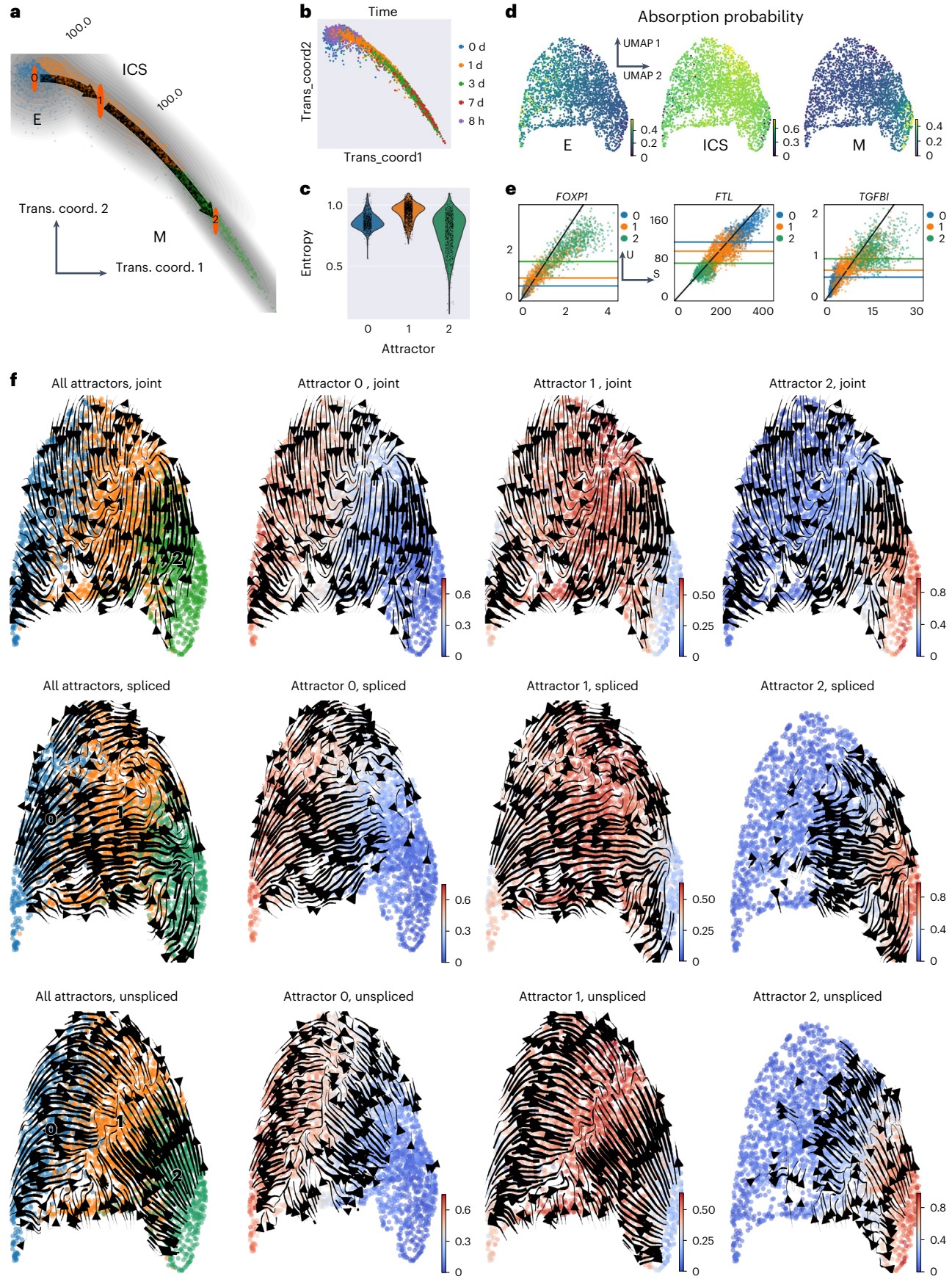

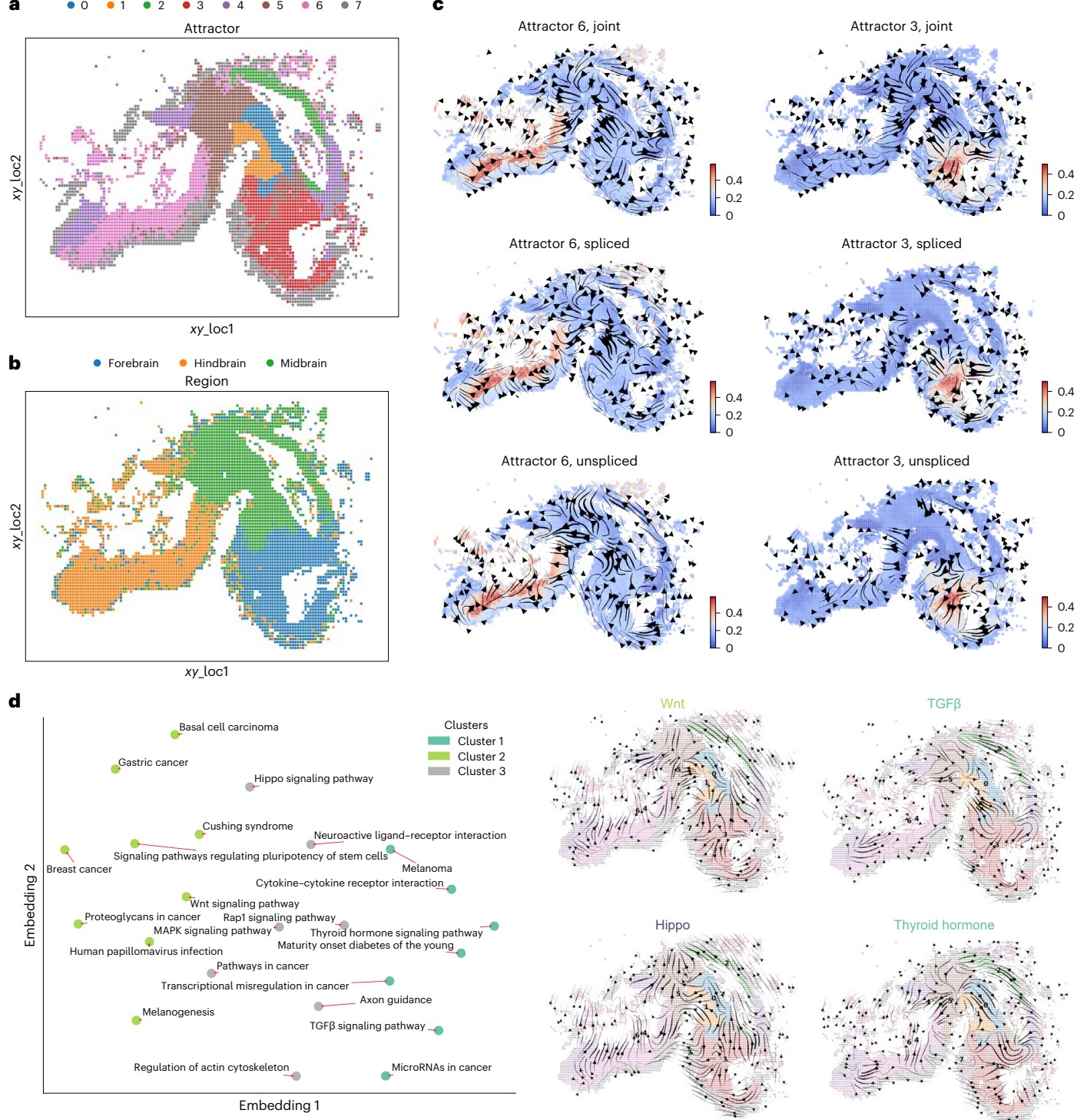

**Fig. 4 | Transition tensor analysis of HybISS mouse brain spatial transcriptomics dataset. a,b,** The spatial annotation of data and detected attractor by STT with cells colored by different categories: attractor (**a**) and region (**b**). **c,** Local transition tensor streamlines in specific attractors 6 and 3. The cells are colored by their memberships in corresponding attractors. **d,** Similarity of transition tensors across KEGG pathways. The left shows 2D embedding indicating the clustering of similar biological pathways in mouse brain development spatial dynamics, with the averaged tensor streamlines from various pathways displaying different transition dynamics. Pathways that have at least three genes overlapped with STT multistability genes are shown in the low-dimensional embeddings. The right shows the streamlines of specific pathways from different clusters, with cells embedded in spatial coordinates.

In addition, we applied STT to blood[41] and pancreas[7] development datasets and found its capability to resolve complex state transitions, and its multistability tensor kernel is consistent with CellRank analysis (Supplementary Figs. 4 and 5).

## STT identifies spatial attractors and pathway similarities

We next applied STT to the HybISS spatial dataset of mouse brain development[42]. To enrich the unspliced and spliced counts for better tensor estimation, we used the SIRV[19] algorithm to impute one of the

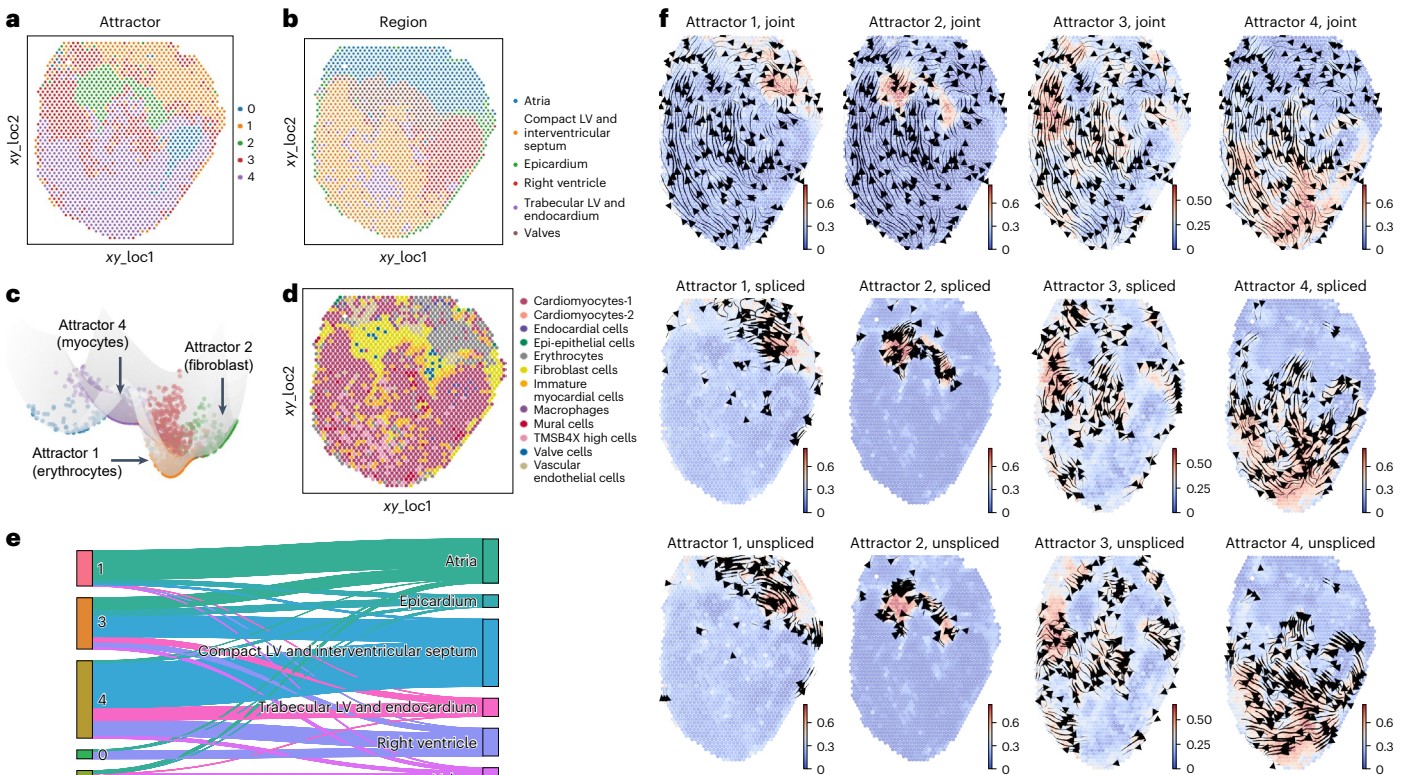

**Fig. 5 | Transition tensor analysis of 10X Visium chicken heart spatial data at day 14. a,b**, The spatial spots of the analyzed data, with spots colored by detected attractor by STT regions (**a**) or annotation in original research (**b**). **c**, The constructed dynamical landscape of data, with spots colored by attractors. **d**, The spatial spots colored by cell type annotations in original research. **e**, The

Sankey plot displaying the relation between STT attractors (left) and spatial region annotations (right). The width of links indicates the number of cells that share the connected attractor label and region annotation label simultaneously. **f**, Local transition tensor streamlines in specific attractors 1, 2, 3 and 4. The cells are colored by their memberships to corresponding attractors.

original spatial data slices at 40 µm at E10 and E11. Compared with clustering only based on cellular similarity (Supplementary Fig. 6), STT identifies attractors consistent with spatial locations of different cell states (Fig. 4a) and brain region annotations in original publication (Fig. 4b): the cells within the same attractor tend to have similar spatial coordinates and belong to the same regions. In addition, the cell assignment is found to be robust to the weight of spatial diffusion kernels (Supplementary Fig. 6), attractor initialization (Supplementary Fig. 7), multistability genes filtering (Supplementary Fig. 8) and number of attractors (Supplementary Fig. 9). The local transition tensors in the forebrain and hindbrain attractors (Fig. 4c) are consistent with UniTVelo analysis (Supplementary Fig. 6).

To evaluate the biological significance of the tensor streamlines, we performed pathway-specific analysis to evaluate functions associated with the cell-state transitions and pathway regulations (Fig. 4d). We used the Kyoto Encyclopedia of Genes and Genomes (KEGG) knowledge database, and calculated the similarity among pathways based on tensor correlations of multistable genes for each pathway (Methods). Indeed, the pathway-specific tensor demonstrates distinct attractor dynamics. The latent embedding and clustering of pathways based on tensor correlation (Fig. 4d) reveal the functional similarity of spatial state transitions between pathways during developmental process. The TGF-beta and WNT pathways, known to exhibit cross-talk and cooperate during embryogenesis[43], are from distinct clusters in the latent embedding, and their tensor streamlines are in opposite directions, especially in midbrain and forebrain attractors (Fig. 4d). Two other important pathways in brain development, the Hippo and Thyroid hormone signaling pathways[44,45], are also from different clusters of pathway tensors, showing opposite streamlines in midbrain and forebrain regions (Fig. 4d). Overall, STT provides dynamical information

for the spatial organizations of cell states and the relations between pathways regulating state transitions during development.

### STT reveals spatial attractors and lineage in chicken heart

We applied STT to the spatially resolved chicken heart data measured by 10X Visium technology[46]. Our analysis is focused on the last temporal point at day 14 from the dataset when the four-chamber development has finalized with completed events of cardiogenesis and explicit spatial boundaries[46].

Using SIRV-imputed unspliced and spliced counts[19], STT identifies five spatially resolved attractors (Fig. 5a and Supplementary Fig. 10). Among them, attractor 2 coincides with the 'valves' region in the original study, and it mainly consists of fibroblast cells (Fig. 5a,b,d,e). Attractor 0 mainly consist of cells from the right ventricle region (Fig. 5e). Attractor 1 mainly localizes in the 'atria' region (Fig. 5e) and is composed of erythrocytes. While the remaining attractors (3, 4) are distributed across several connected regions, they all include the cells of annotated phenotype of cardiomyocytes (Fig. 5a,b,d,e). The dynamical manifold reveals those discrete attractors (Fig. 5c) relate to various cell lineages. The attractors 1 and 2, which contain spatially localized lineages of fibroblasts and erythrocytes, all exhibit the 'attraction force' as seen in the tensor streamlines (Fig. 5f). In comparison, the streamlines of tensors within attractors 3 and 4 (both containing cardiomyocytes) indicate their transience in space and show a tendency to transit into atrial regions, which is also observed in the 'attraction' between unspliced components. This could partly be explained by the existence of another group of myocytes in the atria (Fig. 5b,d). Overall, the observed consistency with spatial regions or cell type annotations indicates STT's capability to dissect spatially resolved attractors.

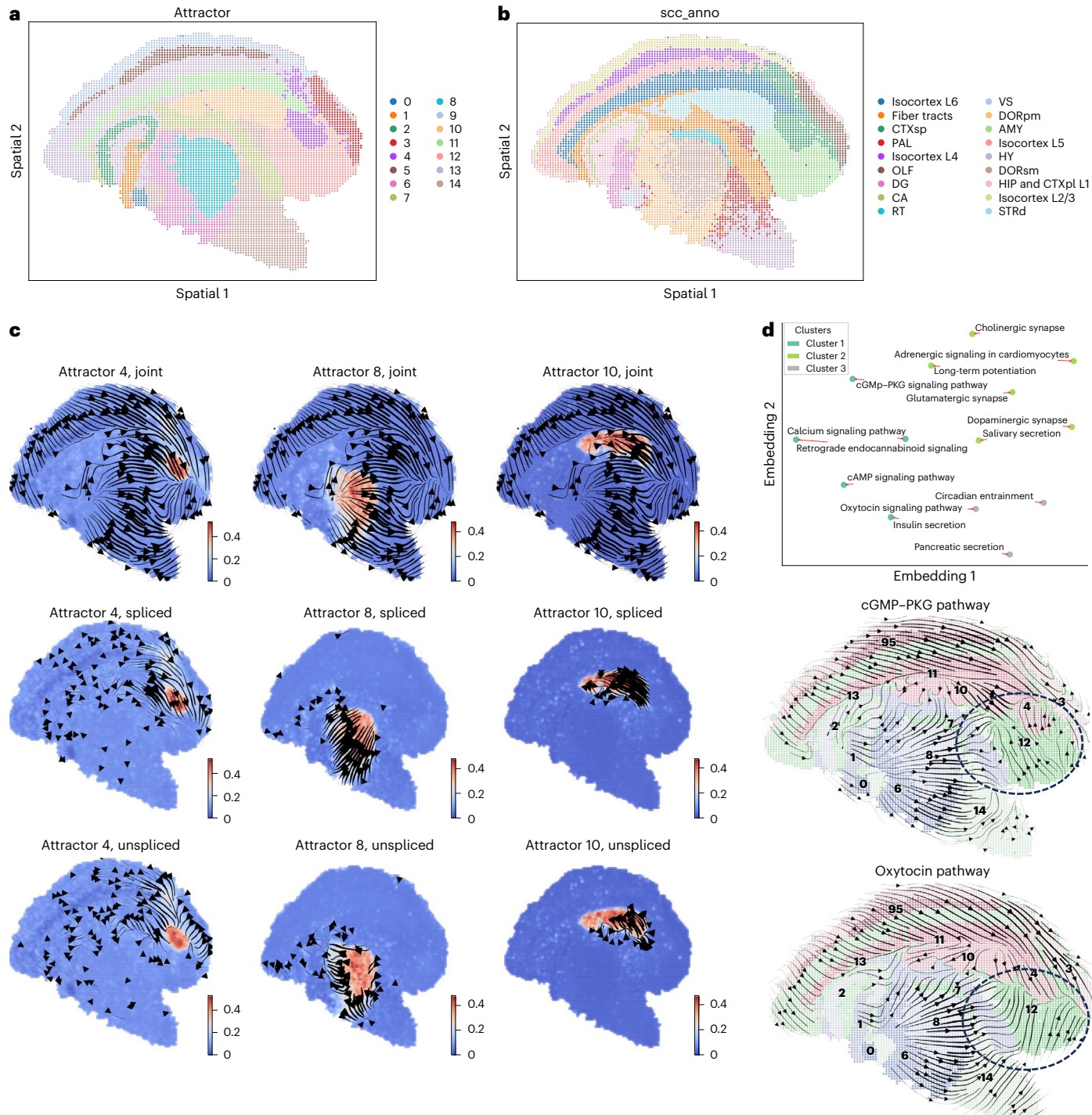

**Fig. 6 | Transition tensor analysis of Stereo-seq mouse coronal hemibrain spatial data. a,b**, The spatial location of cells colored with STT attractors (**a**) and annotation in original research (**b**). **c**, Local streamlines of tensors in attractors 4, 8 and 10. **d**, The 2D embedding of the pathway dynamics (top) and the averaged tensor streamlines of two specific pathways (bottom) with cells colored by attractors and embedded in spatial coordinates. Pathways that have at least eight genes overlapped with STT multistability genes are shown in 2D embedding.

## STT elucidates region-specific spatial attractors and stabilities

We next analyzed the high-resolution Stereo-seq mouse adult coronal hemibrain dataset[47] processed with bin size 60, which revealed the complex domains of neuron cells with various biological functions. Direct application of STT shows several region-specific spatial attractors that are very consistent with the functional annotations of brain regions (Fig. 6a,b). The convergent streamlines of tensors (Fig. 6c)

suggest that the multistability of gene expression dynamics is well maintained in regions such as the cortical subplot (attractor 4) and the striatum dorsal region (attractor 10). Streamlines flow outward (Fig. 6c) in thalamus regions (attractor 8) all tensor components, suggesting its relatively high plasticity. The pathway embedding based on their tensor dynamics showed that the previously known interacting pathways such as cGMP–PKG and the calcium signaling pathway[48] share similar tensor dynamics (Fig. 6d). It also suggests that cGMP–PKG is different

from the oxytocin pathway, in which the streamlines indicate the major differences occurring in the amygdalar nucleus region (Fig. 6d). Overall, the results indicated that STT can discover spatial regions and quantify their stabilities through attractor analysis, even in mature tissues.

## Discussion

Quantifying and modeling the relative abundance between unspliced and spliced counts has enabled an effective mechanistic approach to dissect cell-state transitions from scRNA-seq datasets. To connect the different time scales among gene expression, mRNA splicing and cell lineage dynamics, as well as to study the underlying attractors of these states, we have developed the STT for (1) constructing the attractor-wise transition tensor, (2) analyzing the probabilistic transition paths and transitional cells and (3) inferring the genes that account for the multistability of cell states. This was done through an iterative computation process between (1) parameter inference in transition tensor models and (2) multiscale analysis of tensor-induced stochastic dynamical systems.

Compared with the RNA velocity models, STT is unique in uncovering attractors underlying both the gene expression and the splicing dynamics, as well as quantifying the transitions among them. By assuming multistability, STT is robust to initial state specifications or hidden time correction[7,26,36,49]. The cell-membership functions quantify transitional cells in estimating the transition tensors, naturally bridging the downstream multiscale dynamical analysis.

To identify transitional cells and infer transition paths, STT leverages the computed transition tensor, instead of direct usage of RNA velocity such as CellRank[8] or Dynamo[25]. The multistable transition tensor is found to be more compatible with the attractor assumption in downstream analysis. The iterative scheme of STT between tensor construction and dynamical dissection is found to better ensure such self-consistency. However, since the attractor assumption does not account for oscillation dynamics, STT needs to be improved to capture the nonequilibrium features of datasets with strong cell cycle effects.

The velocity kernel-based cellular random walk derived from the transition tensor is critical for connecting the modules of tensor inference and dynamical decomposition in STT, allowing better-connected dynamics at different scales. Theoretical analysis has revealed that different choices of velocity kernel lead to various continuum limits in forms of ordinary or SDEs[49]. In STT, the inner-product kernel is used to construct the cellular random walk that was shown to be consistent with the stochastic chemical Langevin model of gene expression[49,50], while the cosine kernel, which correctly recovers the directionality of the velocity field[49], is adopted to visualize the local streamlines within attractors. In addition to the differential equation models, it may be interesting to formulate STT in the chemical master equation framework[51] of RNA velocity in the future.

As a mechanistic model-based approach, STT may be improved in several aspects. Instead of using attractor-specific zeroth order approximation of nonlinear gene expression rate function in equation (1), higher-order gene interactions could be considered as proposed recently for gene regulatory network inference[15]. Multimodal information including single-cell epigenomics[52] or proteomics data[53] can also be incorporated in the multistable dynamical system to enhance the transition tensor calculation. The automatic detection of root and target states in multistable models is always challenging, and the previous knowledge or knowing the properties related to cells' differentiation potencies[54,55] could be helpful.

Overall, STT provides a unified approach to extract spatiotemporal information from single-cell datasets by bridging the processes across different time scales and tissue regions. Our method allows for a multiscale description of tissue spatiotemporal structures, connecting microscopic dynamics of gene expression and splicing, and the macroscopic dynamics of cell-state transitions among emergent attractors.

## Online content

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

## Methods

### Multistability in gene expression and splicing model

We use a simple dynamical model with different parameters around each steady-state to approximate the mRNA splicing dynamics for gene $i$:

$$\begin{cases} \frac{dU_i}{dt} = \alpha_{c,i} - \beta_i U_i, \\ \frac{dS_i}{dt} = \beta_i U_i - \gamma_i S_i. \end{cases}$$

Here, $\alpha_{c,i}$ is the state-dependent unspliced mRNA transcription rate in attractor $c$, $\beta_i$ is the mRNA splicing rate and $\gamma_i$ is the mRNA degradation rate. Assuming that the system is close to steady-state, we have $\epsilon_i = \alpha_{c,i} - \beta_i U_i, \eta_i = \beta_i U_i - \gamma_i S_i$ where $\epsilon_i$ and $\eta_i$ are independent and identically distributed zero-mean Gaussian variables. Due to the invariance of scales in parameters[56], we set $\gamma_i = 1$ and the maximum likelihood estimation could be expressed as

$$\min_{\alpha_c, \beta} \sum_{c=1}^{K} \sum_{k=1}^{N_C} (\alpha_c - \beta U_k)^2 1_{k \in \Omega_c} + \sum_{k=1}^{N_C} (\beta U_k - S_k)^2.$$

Since the parameters of different genes are estimated independently, for simplicity of notations, here we omit gene subscript $i$ and introduce the subscript $k$ to denote the cell index. The indicator function of attractors $1_{k \in \Omega_c}$ is initialized using user-provided cell labels or standard Leiden or Louvain clustering algorithm output, and it is updated using membership function in iterations (described below). The estimation yields the solution:

$$\alpha_c^{(*)} = m_c \beta^{(*)}, \beta^{(*)} = \frac{\sum_{k=1}^{N_C} U_k S_k}{\sum_{k=1}^{N_C} \left( U_k^2 + \sum_{c=1}^{K} (U_k - m_c)^2 1_{k \in \Omega_c} \right)}.$$

where $m_c = \frac{\sum_{k=1}^{N_C} U_k 1_{k \in \Omega_c}}{N_c}$. Compared with steady-steady parameter estimation in the standard RNA velocity model, the splicing rate parameter $\beta$ in the multistable model is not only attractor-type specific, but also depends on both unspliced and spliced counts.

For each cell $k$ with counts $(U_k, S_k)$, its velocity with respect to each attractor $c$ is defined as $v_{k,u,c} = \alpha_c^{(*)} - \beta^{(*)} U_k, v_{k,s,c} = \beta^{(*)} U_k - S_k$ where subscript $u$ and $s$ corresponds to unspliced and spliced counts, respectively. This estimation is repeated for each gene, therefore, leading to a four-dimensional transition tensor $v_{k,l,c,g} \in \mathbb{R}^{N_C \times 2 \times K \times N_G}$.

### Tensor-based and spatial-constrained transition dynamics

Next, STT constructs the Markov chain transition probabilities among individual cells based on the calculated tensor, gene expression similarity and spatial coordinates if available (Fig. 1g). The transition probability is constructed from three components: $P = w_1 P^v + w_2 P^c + (1 - w_1 - w_2) P^s$, where $P^v$, $P^c$ and $P^s$ are transition probabilities induced by velocity, similarity and spatial kernels, respectively. Here $w_1$ and $w_2$ are the hyperparameters of the algorithm to balance the effects of different modalities of tensor dynamics, gene expression similarity and spatial closeness. Their effect on output has been tested in Supplementary Fig. 6.

To construct $P^v$, we first transform the attractor-specific tensor to the attractor-independent velocity $V$ by averaging along the dimension of attractors:

$$V_{k,u,g} = \sum_c \rho_{k,c} v_{k,u,c,g}, V_{k,s,g} = \sum_c \rho_{k,c} v_{k,s,c,g}.$$

Here $\rho_{k,c}$ denotes the membership function of cell $k$ in attractor $c$. The stable cell $j$ located around the fixed point of the attractor basin $d$ yield $\rho_{j,d} = 1$, while transitional cell $l$ near saddle points has multiple positive components in $\rho_l$, pointing toward the attractors to which the cell can transition into.

Having calculated the tensor, we next construct the velocity-induced transition probability $P^v$ using the inner-product kernel[49] (Supplementary Note 1). The weight of transition propensity from cell $k$ to $l$ is $w_{kl} = \exp(V_{k,u}^T \Delta U_{kl} + V_{k,s}^T \Delta S_{kl})$ where $\Delta U_{kl} = U_l - U_k$ and $\Delta S_{kl} = S_l - S_k$. The random walk induced by such a kernel is consistent with the SDE model of equation (1) (ref. 49 and Supplementary Note 1). The cell similarity induced transition probability $P^c$ is constructed from the Gaussian kernel of the diffusion map based on gene expression counts[2]. Last, the spatially constrained transition probability $P^s$ is constructed from the Gaussian kernel of spatial location coordinates, such that cells with similar spatial locations are more likely to make transitions between each other. As a result, such cells are more likely to be assigned into the same attractor basins.

To calculate the membership function in attractors, we use the GPPCA[57] algorithm to decompose the constructed random walk transition probability matrix $P$ and coarse-grain the nonequilibrium Markov chains and obtain $\rho_{k,c}$. This algorithm allows for the factoring and 'coarse-graining' of nonequilibrium transition probability matrices of cellular random walk, which holds true for most of the velocity-induced dynamics, to obtain the attractor within the data as well as cell's relevant position (that is, membership) in each attractor. The coarse-grained (cg) transition probability matrix $P_{cg}^{K \times K}$ on the attractor level ($K$ is the total number of attractors) is obtained simultaneously using the GPPCA algorithm. Given the cell's membership function, its transitional entropy can be defined as $\varepsilon_i = - \sum_{c=1}^{K} \rho_{i,c} \ln \rho_{i,c}$. The larger entropy indicates the higher propensity of the cell to make transitions between attractors.

### Iterative scheme for parameter estimation and attractor membership quantification

After obtaining the membership function, the parameters of the tensor model are updated to incorporate the uncertainty of the cells' positions in attractors. We define a loss function

$$\mathcal{J}(\alpha_c, \beta, \rho_{k,c}) = \sum_{c=1}^{K} \sum_{k=1}^{N_C} (\alpha_c - \beta U_k)^2 \rho_{k,c} + \sum_{k=1}^{N_C} (\beta U_k - S_k)^2 + \lambda \sum_{c=1}^{N_C} \alpha_c^2 + \lambda \beta^2,$$

where $\lambda$ denotes the strength of regularization term of kinetic parameters. Intuitively, the 'stable cells' in attractor $c$ have larger weight values in the regression loss function since the confidence level about steady-state is larger. We analytically solve the optimizer

$$\alpha_c^{(*)} = m_c \beta^{(*)}, \beta^{(*)} = \frac{\sum_{k=1}^{N_C} U_k S_k}{\sum_{k=1}^{N_C} \left( U_k^2 + \sum_{c=1}^{K} (U_k - m_c)^2 \rho_{k,c} \right) + \lambda},$$

where $m_c = \frac{\sum_{k=1}^{N_C} U_k \rho_{k,c}}{\sum_{k=1}^{N_C} \rho_{k,c} + \lambda}$.

In turn, the updated tensor with the newly optimized parameters leads to an updated membership function. In STT, we adopt an iterative scheme to update tensor parameters and attractor memberships jointly,

$$\alpha_c^{n+1}, \beta^{n+1} = \operatorname{argmin}_{\alpha_c, \beta} \mathcal{J}\left(\alpha_c, \beta, \rho_{k,c}^n\right),$$

$$\rho_{k,c}^{n+1} = \text{DynamicalAnalysis}(\alpha_c^n, \beta^n)$$

where the superscript $n$ denotes the number of iterations, and DynamicalAnalysis denotes the described procedure to update membership function. The scheme stops once the membership function does not improve within certain threshold, or the iteration exceeds the allowed maximum number of iterations.

To dissect the multistable dynamics accurately, we also filter the genes in each iteration based on their goodness of fit to model that includes the genes showing multistability. The metric of goodness, or gene multistability score, is defined as $1 - \frac{\mathcal{J}(\alpha_c, \beta, \rho_{k,c})}{N_C (\text{Var}(U) + \text{Var}(S))}$ where $N_C$

denotes the number of cells. Only the tensor of filtered genes, whose multistability scores are larger than certain threshold, are used to calculate velocity kernel and therefore update the membership function. The hyperparameters of STT and their values chosen in datasets analyzed are presented in Supplementary Tables 1 and 2.

To allow robust control of the iteration scheme, we incorporated a monitor module that outputs the multistability scores of genes in both training (by default 80% of all sample sizes) and test dataset (20% of all samples). The training dataset was used to fit kinetic parameters $(\alpha_c, \beta)$, and the multistability scores of genes are calculated on test dataset. The monitor module outputs the multistability scores and number of genes pass the threshold. Given the output, the user may choose to interactively (1) modify the threshold set for filtering multistability genes, (2) adjust the weight of tensor-induced kernel against gene expression similarity or spatial kernels to encourage the high-quality transition matrices or (3) determine whether to stop the iteration, therefore facilitating the adaptive accuracy. The interface of monitor module is demonstrated in Supplementary Fig. 1f. We also demonstrate the efficiency and scalability of STT algorithm in Supplementary Table 3 and Supplementary Fig. 10.

### Initialization of iteration
To start the iteration, STT requests the input of existing clustering results to create attractor membership by one-hot encoding. The previous biological annotation of the dataset or spatial region segmentation results were recommended as the input. When such information is unavailable, users may adopt clustering algorithms such as Leiden or Louvain to cluster the cells based on expression counts (spliced only or spliced and unspliced jointly) or spatial location of the cells. The robustness to initialization of STT was investigated (Supplementary Fig. 7). Whenever the user prefers alternative clustering methods and/or more systematic analysis, STT provides an option to feed a user-generated clustering output as the input for the initializations of STT.

### Visualization of dynamical manifold and transition paths
To visualize the low-dimensional embeddings of cells, STT uses the join state $x_k = (U_k, S_k) \in \mathbb{R}^{2N_G}$ for each cell $k$ as the input of dimensionality reduction algorithms such as principal component analysis (PCA) or uniform manifold approximation and projection (UMAP). To visualize the dynamical manifold, we define the cell's position in the two-dimensional (2D) plane as $y_k = \sum_{c=1}^{K} \rho_{k,c}\mu_c$, where $\mu_c$ is the center of PCA or UMAP embeddings of each attractor and $K$ is the number of attractors. Then, a Gaussian mixture density estimation $\mathcal{P}(y)$ is constructed for all $y_k$ using an expectation–maximization algorithm, where the initial weights for $K$ components are the stationary distributions of attractor-level, coarse-grained random walk transition probability matrix $P$ derived in the previous section. The surface of dynamical manifold was calculated as $\phi(y) = -\ln\mathcal{P}(y)$. The streamlines of the velocity $V_k = (V_{k,u}, V_{k,s}) \in \mathbb{R}^{2N_G}$ in the 2D plane are calculated using the linear (PCA) or nonlinear (UMAP) projection approach in scVelo with the cosine kernel. Given initial and final states, the transition paths and their proportion of the total transition probability flux are calculated using the transition path theory[58] with the PyEMMA package[59].

### Synthetic datasets and benchmarking
The simulation data ($n = 10,010$ cells) for the toggle-switch system were generated by the SDE model of a mutually inhibited two-gene circuits with nonlinear gene regulation and/or splicing dynamics and stochastic noise (Supplementary Note 2). Two attractors are present in the system with a saddle point in between. The synthetic dataset ($n = 5,000$ cells) of EMTs was generated by the SDE model of a seven-gene core circuit during EMT adapted to include mRNA splicing[15,60]. With different levels of extrinsic signal TGFB, the system has saddle-node bifurcations within a certain parameter range and three attractors may coexist, representing epithelial state, ICS and mesenchymal state (Supplementary Note 2).

We simulated different levels of TGFB to model the EMT process. For both datasets, the Euler–Maruyama method was used to simulate the SDE trajectories, with negative gene expression values adjusted to zero during each time step of the trajectory simulation.

### Pathway analysis
To analyze the similarities between tensor dynamics in various pathways, we first downloaded the pathway databases, such as KEGG, using the GSEApy package[61]. Next, for each pathway we identified the genes shared by the pathway databases and the STT multistability analysis. For any selected gene sets that contain a sufficient number of genes, we calculated their cosine-kernel velocity graph using the averaged tensor of both spliced counts and unspliced counts, and then computed the Pearson's correlation coefficients between pathway-specific velocity graphs. The UMAP dimensionality reduction of pathways was then performed on the principal components of the correlation matrix, and clustering was performed on UMAP with $K$-means algorithm by silhouette score to choose the optimal number of clusters.

### Reporting summary
Further information on research design is available in the Nature Portfolio Reporting Summary linked to this article.

### Data availability
All the datasets used in this paper are publicly available. The detailed preprocessing of datasets is described in Supplementary Notes. The simulation datasets of synthetic circuits are available at https://github.com/cliffzhou92/STT/tree/release/data. The EMT dataset of human lung A549 cell lines is available at GSE147405. The pancreas dataset (originally available at GSE132188) and adult human bone marrow datasets (originally available at https://data.humancellatlas.org/explore/projects/091cf39b-01bc-42e5-9437-f419a66c8a45) can be downloaded from the built-in datasets of the scvelo==0.2.4 package (https://scvelo.readthedocs.io/en/stable/api.html). The spatial datasets of mouse brain and chicken heart as well as scRNA-seq datasets used for imputation can be downloaded from https://doi.org/10.5281/zenodo.6798658 (ref. 62). The Stereo-seq mouse brain dataset with unspliced and spliced counts was downloaded from the Spateo package (https://github.com/aristoteleo/spateo-tutorials, https://www.dropbox.com/s/c5tu4drxda01m0u/mousebrain_bin60.h5ad?dl=0). The KEGG database was originally available on the Enrichr webpage (https://maayanlab.cloud/Enrichr/#librariesdownloaded) and downloaded using gseapy==1.0.4. The processed datasets for analysis are also stored at https://disk.pku.edu.cn/link/AAD1681DAD531D47699D459BB46C4651D8.

### Code availability
STT is implemented as a Python package available at https://github.com/cliffzhou92/STT/tree/release. The source code for simulation and the notebook files to reproduce all analysis in the paper are also available at https://github.com/cliffzhou92/STT/tree/release/example_notebooks.

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

## Acknowledgements

This project was supported by National Institutes of Health grant nos. U01AR073159 and R01AR079150 (Q.N.), National Science Foundation grant nos. DMS1763272 (Q.N.) and MCB2028424 (Q.N.), The Simons Foundation (grant no. 594598 to Q.N.), NSFC grants no. 11825102 (T.L.), no. 12288101 (T.L. and P.Z.) and no. 8206100646 (P.Z.), National Key R&D Program of China grant no. 2021YFA1003300 (T.L.), Start-up grants of Peking University grant no. 7101303365 (P.Z.) and the High-performance Computing Platform of Peking University (P.Z.).

## Author contributions

Q.N. and P.Z. conceived the project. P.Z. and T.L. designed the algorithm and implemented the code. P.Z. and F.B. performed the simulations, analyzed the data and interpreted the results. Q.N. and P.Z. wrote the paper. All authors revised and approved the paper. Q.N. supervised the project.

## Competing interests

The authors declare no competing interests.

## Additional information

**Extended data** is available for this paper at https://doi.org/10.1038/s41592-024-02266-x.

**Correspondence and requests for materials** should be addressed to Qing Nie.

# Reporting Summary

## Statistics

For all statistical analyses, confirm that the following items are present in the figure legend, table legend, main text, or Methods section.

| n/a | Confirmed | |
|---|---|---|
| ☐ | ☒ | The exact sample size ($n$) for each experimental group/condition, given as a discrete number and unit of measurement |
| ☐ | ☒ | A statement on whether measurements were taken from distinct samples or whether the same sample was measured repeatedly |
| ☐ | ☒ | The statistical test(s) used AND whether they are one- or two-sided<br>*Only common tests should be described solely by name; describe more complex techniques in the Methods section.* |
| ☒ | ☐ | A description of all covariates tested |
| ☒ | ☐ | A description of any assumptions or corrections, such as tests of normality and adjustment for multiple comparisons |
| ☒ | ☐ | A full description of the statistical parameters including central tendency (e.g. means) or other basic estimates (e.g. regression coefficient) AND variation (e.g. standard deviation) or associated estimates of uncertainty (e.g. confidence intervals) |
| ☒ | ☐ | For null hypothesis testing, the test statistic (e.g. $F$, $t$, $r$) with confidence intervals, effect sizes, degrees of freedom and $P$ value noted<br>*Give P values as exact values whenever suitable.* |
| ☒ | ☐ | For Bayesian analysis, information on the choice of priors and Markov chain Monte Carlo settings |
| ☒ | ☐ | For hierarchical and complex designs, identification of the appropriate level for tests and full reporting of outcomes |
| ☒ | ☐ | Estimates of effect sizes (e.g. Cohen's $d$, Pearson's $r$), indicating how they were calculated |

*Our web collection on statistics for biologists contains articles on many of the points above.*

## Software and code

Policy information about availability of computer code

| Data collection | The bone marrow dataset and pancreas dataset was downloaded using cellrank (1.3.1) and scvelo (0.2.4) package. The mouse brain and checken heart data was provided with package SIRV 1.0 (https://github.com/tabdelaal/SIRV) with datasets deposited on zenodo with doi https://doi.org/10.5281/zenodo.6798659. The KEGG database was downloaded using gseapy==1.0.4 |
|---|---|
| Data analysis | The STT software is available at  and its dependency on other  packages was described in the file. The software described in manuscript includes cellrank(1.3.1)which implements GPCCA algorithm, and pyEMMA(2.5.6) which implements transition paths analysis. |

For manuscripts utilizing custom algorithms or software that are central to the research but not yet described in published literature, software must be made available to editors and reviewers. We strongly encourage code deposition in a community repository (e.g. GitHub). See the Nature Portfolio guidelines for submitting code & software for further information.

## Data

Policy information about availability of data

All manuscripts must include a data availability statement. This statement should provide the following information, where applicable:
- Accession codes, unique identifiers, or web links for publicly available datasets
- A description of any restrictions on data availability
- For clinical datasets or third party data, please ensure that the statement adheres to our policy

All the datasets used in this paper are publicly available. The detailed preprocessing of datasets is described in Supplementary Note. The simulation datasets of

# Research involving human participants, their data, or biological material

Policy information about studies with human participants or human data. See also policy information about sex, gender (identity/presentation), and sexual orientation and race, ethnicity and racism.

| | |
|---|---|
| Reporting on sex and gender | n/a, this study does not involve human participants. |
| Reporting on race, ethnicity, or other socially relevant groupings | n/a, this study does not involve human participants. |
| Population characteristics | n/a, this study does not involve human participants. |
| Recruitment | n/a, this study does not involve human participants. |
| Ethics oversight | n/a, this study does not involve human participants. |

Note that full information on the approval of the study protocol must also be provided in the manuscript.

# Field-specific reporting

Please select the one below that is the best fit for your research. If you are not sure, read the appropriate sections before making your selection.

☒ Life sciences    ☐ Behavioural & social sciences    ☐ Ecological, evolutionary & environmental sciences

For a reference copy of the document with all sections, see nature.com/documents/nr-reporting-summary-flat.pdf

# Life sciences study design

All studies must disclose on these points even when the disclosure is negative.

| | |
|---|---|
| Sample size | To evaluate the performance of STT under various scenarios, we tested on two simulation datasets, three scRNA-seq datasets and three spatial transcriptome datasets sequenced by different technology |
| Data exclusions | No data was excluded |
| Replication | We run the analysis code multiple times to ensure replication of results |
| Randomization | Randomization is not relevant to our study  since we do not perform group allocation and comparison |
| Blinding | Blinding is not relevant to this study since we do not perform group allocation and comparison |

# Reporting for specific materials, systems and methods

We require information from authors about some types of materials, experimental systems and methods used in many studies. Here, indicate whether each material, system or method listed is relevant to your study. If you are not sure if a list item applies to your research, read the appropriate section before selecting a response.

## Materials & experimental systems

| n/a | Involved in the study |
|-----|----------------------|
| ☒ ☐ | Antibodies |
| ☒ ☐ | Eukaryotic cell lines |
| ☒ ☐ | Palaeontology and archaeology |
| ☒ ☐ | Animals and other organisms |
| ☒ ☐ | Clinical data |
| ☒ ☐ | Dual use research of concern |
| ☒ ☐ | Plants |

## Methods

| n/a | Involved in the study |
|-----|----------------------|
| ☒ ☐ | ChIP-seq |
| ☒ ☐ | Flow cytometry |
| ☒ ☐ | MRI-based neuroimaging |

## Plants

| | |
|---|---|
| Seed stocks | *Report on the source of all seed stocks or other plant material used. If applicable, state the seed stock centre and catalogue number. If plant specimens were collected from the field, describe the collection location, date and sampling procedures.* |
| Novel plant genotypes | *Describe the methods by which all novel plant genotypes were produced. This includes those generated by transgenic approaches, gene editing, chemical/radiation-based mutagenesis and hybridization. For transgenic lines, describe the transformation method, the number of independent lines analyzed and the generation upon which experiments were performed. For gene-edited lines, describe the editor used, the endogenous sequence targeted for editing, the targeting guide RNA sequence (if applicable) and how the editor was applied.* |
| Authentication | *Describe any authentication procedures for each seed stock used or novel genotype generated. Describe any experiments used to assess the effect of a mutation and, where applicable, how potential secondary effects (e.g. second site T-DNA insertions, mosiacism, off-target gene editing) were examined.* |

