## [Peer Review File · Nature Methods]

Peer Review Information

Manuscript Title: Spatial Transition Tensor of Single Cells

Corresponding author name(s): Qing Nie

Editorial Notes: None

Reviewer Comments & Decisions:

Decision Letter, initial version:

Dear Qing,

Your Article, "Spatial Transition Tensor of Single Cells", has now been seen by 2 reviewers. As you will see from their comments below, although the reviewers find your work of considerable potential interest, they have raised a number of concerns. We are interested in the possibility of publishing your paper in Nature Methods, but would like to consider your response to these concerns before we reach a final decision on publication.

We therefore invite you to revise your manuscript to address these concerns particularly concerns about lack of benchmarking and concerns about reproducibility.

[Redacted]

We hope to receive your revised paper within eight weeks. If you cannot send it within this time, please let us know. In this event, we will still be happy to reconsider your paper at a later date so long as nothing similar has been accepted for publication at Nature Methods or published elsewhere.

OPEN SCIENCE REQUIREMENTS

REPORTING SUMMARY AND EDITORIAL POLICY CHECKLISTS

DATA AVAILABILITY

We strongly encourage you to deposit all new data associated with the paper in a persistent repository where they can be freely and enduringly accessed. We recommend submitting the data to discipline-specific and community-recognized repositories; a list of repositories is provided here:

<http://www.nature.com/sdata/policies/repositories>

All novel DNA and RNA sequencing data, protein sequences, genetic polymorphisms, linked genotype and phenotype data, gene expression data, macromolecular structures, and proteomics data must be deposited in a publicly accessible database, and accession codes and associated hyperlinks must be provided in the "Data Availability" section.

CODE AVAILABILITY

Please include a "Code Availability" subsection in the Online Methods which details how your custom code is made available. Only in rare cases (where code is not central to the main conclusions of the paper) is the statement "available upon request" allowed (and reasons should be specified).

For more information on our code sharing policy and requirements, please see: <https://www.nature.com/nature-research/editorial-policies/reporting-standards#availability-of-computer-code>

MATERIALS AVAILABILITY

SUPPLEMENTARY PROTOCOL

To help facilitate reproducibility and uptake of your method, we ask you to prepare a step-by-step Supplementary Protocol for the method described in this paper. We encourage authors to share their step-by-step experimental protocols on a protocol sharing platform of their choice and report the protocol DOI in the reference list. Nature Portfolio 's Protocol Exchange is a free-to-use and open resource for protocols; protocols deposited in Protocol Exchange are citable and can be linked from the published article. More details can found at www.nature.com/protocolexchange/about.

ORCID

Sincerely,
Madhura

Madhura Mukhopadhyay, PhD
Senior Editor
Nature Methods

Reviewers' Comments:

Reviewer #1:

Remarks to the Author:

The manuscript presents the Spatial Transition Tensor (STT) method, an innovative approach tailored for the analysis of spatial transcriptomics data. Central to the STT method is a four-dimensional transition tensor, meticulously designed to capture the nuanced dynamics of mRNA splicing in conjunction with spatial transcriptomes. This tensor-based methodology provides a panoramic view of single-cell transcriptome data, encompassing a diverse range of spatiotemporal scales. One of the standout features of the STT method is its ability to identify multiple stable states or "attractors" in the data, which can be indicative of distinct cellular states or phenotypes. The method's iterative process, which alternates between tensor model construction and dynamical dissection, ensures a heightened level of self-consistency in the analysis. This iterative refinement is geared towards

capturing the underlying dynamics with increased precision.

Given the rapidly growing importance of spatial transcriptomics in the realm of cellular dynamics within tissues, the STT method's detailed approach is timely and holds immense potential. Its capability to unravel intricate cellular dynamics, which might be glossed over by simpler models like RNA velocity, is commendable. This depth of analysis can provide transformative insights into cellular transitions, differentiation pathways, and spatial patterns, potentially setting new standards in the interpretation of spatial transcriptomics data.

However, while the transition tensor is undeniably a groundbreaking innovation, there are significant concerns that need addressing, particularly regarding the methodology, its real-world application, and benchmarking.

Major Concern 1: The dynamics between spliced and unspliced mRNA counts are foundational to both the STT method and traditional RNA velocity techniques. These dynamics are pivotal in providing insights into the current cellular state and its future trajectory. However, the inherent assumption of a consistent relationship between spliced and unspliced counts can be a potential pitfall. In biological scenarios where these dynamics deviate from the norm, the STT method's unwavering adherence to this assumption could lead to skewed interpretations of cellular states and trajectories. While many methods, including RNA velocity ones like scvelo, adopt this assumption, the STT method's intricate tensor-based approach and multi-scale perspective might amplify any inaccuracies stemming from this assumption. The spatial component of the data, combined with the STT method's focus on spatial transcriptomics, adds another layer of complexity. To fortify against this potential pitfall, the authors could integrate a dedicated module within the STT framework. This module, designed to continuously monitor the observed dynamics between spliced and unspliced counts, could recalibrate model parameters in real-time, ensuring adaptive accuracy.

Major Concern 2: The STT method's inherent complexity, stemming from its intricate tensor-based approach and iterative scheme, while enabling it to capture nuanced cellular dynamics, also exposes it to the risk of overfitting. This is especially concerning when dealing with smaller datasets or datasets with pronounced variability. Overfitting could distort the biological signal, leading to misleading interpretations. Regularization techniques, such as L1 or L2 regularization, during the parameter estimation phase could mitigate this. Implementing a cross-validation scheme and an early stopping mechanism in the STT method's iterative scheme can further bolster against overfitting. Noise filtering techniques applied before data input can also enhance the method's robustness.

Major Concern 3: The four-dimensional nature of the transition tensor, while innovative, poses interpretability challenges. The authors have commendably employed UMAP for low-dimensional embeddings, but there's potential for further refinement. Focused visualizations highlighting specific tensor features or dynamics, interactive tensor exploration techniques, and detailed summary metrics derived from the tensor could make the STT method more accessible and user-friendly.

Major Concern 4: The STT method's iterative nature might make it sensitive to initial conditions or parameter estimates. This could challenge result reproducibility. Exploring multiple initialization strategies and incorporating mechanisms within the STT framework that automatically adjust initial conditions based on observed data patterns can enhance the method's robustness.

Minor Comments:

(1) Clarity in Method Description: Some sections could benefit from more detailed explanations, especially concerning the mathematical underpinnings of the transition tensor.

(2) Validation on Diverse Datasets: A broader range of datasets could provide a more comprehensive performance assessment.

(3) Comparison with Existing Methods: A detailed comparison, especially in terms of accuracy, computational efficiency, and scalability, would be insightful.

(4) Parameter Sensitivity Analysis: A sensitivity analysis would offer insights into the method's robustness.

(5) User-Friendly Implementation: The provided GitHub page is a step in the right direction, but there's scope for improvement. Annotated Jupyter notebooks, detailed function descriptions, and interactive tutorials could make the STT method more user-friendly.

In summation, the manuscript showcases a promising avenue for spatial transcriptomics data analysis. While the method is poised for significant impact, addressing the highlighted concerns can further refine its robustness and applicability.

Reviewer #3:

Remarks to the Author:

A. Summary of the key results

This work proposes a computational framework called Spatial Transition Tensor (STT) which extends single cell RNA dynamics into multi-scale, cell-state dependent scenarios, and, with the availability of spatial constraints, it allows for modeling spatial transcriptome datasets as well. Instead of a global equilibrium state assumed by the original single cell RNA velocity method, STT assigns cells into a prespecified number of cell clusters, or attractors, where multiple local, cluster-dependent steady states are assumed to be reached. Such treatment allows the derivation of a gaussian likelihood objective which can be solved analytically. To train STT, two major steps are iterated until cell assignment results converge. Initialized with external cell clusters, in step one STT estimates the rate parameters of the dynamic model from data, before cluster-dependent velocities together with the inter-cellular transition probabilities are estimated. In step two, the dynamic transition matrix based on 1) unspliced and spliced velocities, 2) expression similarities; and 3) an optional spatial kernel for spatial transcriptome is decomposed via spectral clustering method, where the membership of each cell to a cluster is derived. Compared with the original RNA velocity method, there is no need of selecting lower and upper quantile cells for approximating steady-state cells. However, a filtering scheme is needed in each iteration for removing genes that can not well fit the data.

B. Originality and significance

Overall, this work is innovative and well-motivated by dynamical systems theory. Conceptually, It contributes to single cell dynamics/trajectory analysis with a more flexible theoretical perspective which is free from the strong assumption made in the original RNA velocity theory; practically, it proposes approximation solutions for model implementation and algorithms for fitting and analysis. Given these contributions, it would be a good resource in the field.

However, although STT is based on dynamical systems theory, the current work did not move further to the validation of its fundamental assumptions on single-cell datasets. As a result, like prior

methods, there may still be scenarios where the method falls short, yet we do not have a tool for relevant diagnosis. Nevertheless, the established fundamental conceptual framework as well as the empirical demonstrations are still valuable references which paved the way for future research.

F. Suggested improvements

For the manuscript, I have the following comments regarding readability, clarity of plotting, additional benchmark experiments, and hyper-parameter sensitivity analysis.

Introduction

1. Perhaps the underlying dynamics model could be introduced in a more intuitive and self-contained manner, with the assistance of concise cartoons or toy examples grounded in single cell biology, so that non-technical readers can quickly catch the point. The current text may assume prior knowledge of dynamic systems, including concepts like “attractors”, “saddle points” as well as the way they are involved in a system.

2. Fig 1.a. and b. seem not very straightforward to understand, without better annotations in the figures or descriptions in the caption/main texts. For example, what do the horizontal lines mean in subfigure STT (Multistable); the visualisation of the transition tensor seems confusing at first glance.

Results:

1. General plotting:

The figures need to be improved by providing more helpful visual indicators on the focused patterns, for example, it would be better to have the ground-truth transition directions marked in simulation (Fig. 2) result, and the patterns of interest highlighted in Fig 3-6, especially when cross-figure comparison is needed.

The arrows look very messy in sub-figures Fig.2b, Fig.4h, k, Fig.6 a-b, Fig.7 f.e, it is sometimes confusing which areas to attend to and compare with.

Cell embeddings are better aligned across different subfigures in comparison, and colored consistently. For example, in Fig 2.a, the colors are not aligned among STT, scVelo UnitVelo plots, and the Dynamo seems to be plotted in a totally different embedding space. The mismatch makes a lot of cognitive load to compare different methods on the same dataset.

2. When doing transition matrix decomposition, what is the motivation of using GPPCA and how is it advantageous over other methods? Given that this is a less popularly used method, it may be helpful to introduce it in more detail.

3. There are multiple hyper-parameters involved in STT modeling, but their effects are not reported:

1) A very important hyper-parameter is the number of attractors, or cell clusters, as attractors decide the course-level trajectory as well as local velocities of cells. In empirical analysis, how would the results change with different numbers of clusters, and with different cluster annotations?

2) Weighting of transition matrices for decomposition. How would different weighting schemes affect model learning?

3) Threshold for gene filtering in the iterative model fitting procedure. Empirically, scVelo can produce different results if different sets of velocity genes are selected for cellular transition estimation.

For clarity, there may better be a summary report of all the hyper-parameters applied in different benchmark datasets.

4. In line 70- 73, and also line 311-314 [Discussion], there is a claim on the limitation of CellRank and

Dynamo based on linear RNA velocity kernels. Concerning this, it can be interesting to also include benchmark datasets on which prior methods were validated, for example, CellRank has included in its package some nice datasets used in their report, e.g., Pancreas and Reprogramming datasets, which are available in the CellRank package <https://cellrank.readthedocs.io/en/latest/api/datasets.html>. It would be nice if STT can show comparable results on these datasets. From another perspective, it might also be interesting to investigate if, based on a multi-stability velocity kernel (e.g. generated by STT), CellRank can improve its performance.

Typos:

Line 358: "... and the most likelihood estimation ..." should be maximum likelihood estimation

Line 394: in the equation, the subscript "i" is most likely "k".

Author Rebuttal to Initial comments

Reviewers' Comments:

Reviewer #1:

Remarks to the Author:

The manuscript presents the Spatial Transition Tensor (STT) method, an innovative approach tailored for the analysis of spatial transcriptomics data. Central to the STT method is a four-dimensional transition tensor, meticulously designed to capture the nuanced dynamics of mRNA splicing in conjunction with spatial transcriptomes. This tensor-based methodology provides a panoramic view of single-cell transcriptome data, encompassing a diverse range of spatiotemporal scales. One of the standout features of the STT method is its ability to identify multiple stable states or "attractors" in the data, which can be indicative of distinct cellular states or phenotypes. The method's iterative process, which alternates between tensor model construction and dynamical dissection, ensures a heightened level of self-consistency in the analysis. This iterative refinement is geared towards capturing the underlying dynamics with increased precision.

Given the rapidly growing importance of spatial transcriptomics in the realm of cellular dynamics within tissues, the STT method's detailed approach is timely and holds immense potential. Its capability to unravel intricate cellular dynamics, which might be glossed over by simpler models like RNA velocity, is commendable. This depth of analysis can provide transformative insights into cellular transitions, differentiation pathways, and spatial patterns, potentially setting new standards in the interpretation of spatial transcriptomics data.

However, while the transition tensor is undeniably a groundbreaking innovation, there are significant concerns that need addressing, particularly regarding the methodology, its real-world application, and benchmarking.

Response: We appreciate the reviewer's careful reading and insightful suggestion of our manuscript. We are grateful for the positive comments on our method. In the revised manuscript, we have made substantial improvements based on the reviewer's suggestions shown as below.

Major Concern 1: The dynamics between spliced and unspliced mRNA counts are foundational to both the STT method and traditional RNA velocity techniques. These dynamics are pivotal in providing insights into the current cellular state and its future trajectory. However, the inherent assumption of a consistent relationship between spliced and unspliced counts can be a potential pitfall. In biological scenarios where these dynamics deviate from the norm, the STT method's unwavering adherence to this assumption could lead to skewed interpretations of cellular states and trajectories. While many methods, including RNA velocity ones like scvelo, adopt this assumption, the STT

method's intricate tensor-based approach and multi-scale perspective might amplify any inaccuracies stemming from this assumption. The spatial component of the data, combined with the STT method's focus on spatial transcriptomics, adds another layer of complexity. To fortify against this potential pitfall, the authors could integrate a dedicated module within the STT framework. This module, designed to continuously monitor the observed dynamics between spliced and unspliced counts, could recalibrate model parameters in real-time, ensuring adaptive accuracy.

Response: Thank you for the insightful suggestion. We agree that a monitor module during iteration process will significantly improve the robustness and flexibility of STT algorithm. In the revised STT, we added a monitor mode that in each step outputs 1) the goodness of fit between spliced and unspliced counts in all genes, 2) the number of genes passing certain user-defined multi-stability score, and 3) the quantiles statistics on absolute or relative differences cells' attractor membership between consecutive iterations. Given the output, the user may choose to interactively 1) modify the threshold set for filtering multi-stability genes, 2) adjust the weight of tensor-induced kernel against gene expression similarity or spatial kernels to encourage the high-quality transition matrices, or 3) determine whether to stop the iteration, therefore facilitating the adaptive accuracy.

In the revised Methods, we added a description of the monitor module (line 458-467). We also demonstrated the user interface of the monitor module in Supplementary Figure S1f. The interactive user interface is also showcased and explained in the Jupyter notebook tutorials for the various datasets.

Major Concern 2: The STT method's inherent complexity, stemming from its intricate tensor-based approach and iterative scheme, while enabling it to capture nuanced cellular dynamics, also exposes it to the risk of overfitting. This is especially concerning when dealing with smaller datasets or datasets with pronounced variability. Overfitting could distort the biological signal, leading to misleading interpretations. Regularization techniques, such as L1 or L2 regularization, during the parameter estimation phase could mitigate this. Implementing a cross-validation scheme and an early stopping mechanism in the STT method's iterative scheme can further bolster against overfitting. Noise filtering techniques applied before data input can also enhance the method's robustness.

Response: Thanks for the suggestion. We agree that using regularization, cross-validation and early stopping could avoid overfitting of STT. Following your suggestion, we have adopted several strategies to improve the robustness of STT:

1) In the tensor construction, we splitted both the spliced and unspliced counts

into a training and a test dataset, and used r^2 metric in the test dataset to filter multi-stability genes.

2) In kinetic parameter estimation, we added the L2 regularization term and made the regularization strength parameter as an optional parameter of STT. We also filter cells with spliced and unspliced counts within 10%-90% quantile in each attractor for more robust performance.

3) The summary statistics was shown (outputted) in each iteration, allowing the user to determine whether to terminate the iteration early (as discussed in detail in response to major concern 1).

4) We studied the robustness of the modified algorithm by subsampling datasets with varied sizes (Figure S1c, S9a).

We have described the strategies to improve STT robustness in the revised Method together with the description of monitor module (line 458-467).

Major Concern 3: The four-dimensional nature of the transition tensor, while innovative, poses interpretability challenges. The authors have commendably employed UMAP for low-dimensional embeddings, but there's potential for further refinement. Focused visualizations highlighting specific tensor features or dynamics, interactive tensor exploration techniques, and detailed summary metrics derived from the tensor could make the STT method more accessible and user-friendly.

Response: Thank you for pointing out this. In the revision, we have adopted several ways to enhance the interpretability of the transition tensors:

- 1) We have used heatmaps to visualize the tensor across different axis, and found that their overall relation is consistent (Fig. S3)
- 2) We have refined the STT plotting module to enable users to plot the tensor dynamics with assigned component, with examples shown in main text Fig. 4c, 5f and 6c.
- 3) In the improved STT output, we also have included summary statistics of the tensor, such as average speed across attractors, genes or cells.

Major Concern 4: The STT method's iterative nature might make it sensitive to initial conditions or parameter estimates. This could challenge result reproducibility. Exploring multiple initialization strategies and incorporating mechanisms within the STT framework that automatically adjust initial conditions based on observed data patterns can enhance the method's robustness.

Response: Thank you for the comment. We agree that the initial condition is important for the STT iteration scheme. In the implementation of STT and tutorial, we allow the users to choose different strategies either based on 1) clustering on spliced counts only; 2) clustering based on both unspliced and

spliced counts; or 3) cell annotations based on prior biology, and the monitor module will output diagnostic statistics for each choice.

In the new benchmarking test, we found that with different initialization strategies, although the constructed tensor might be quantitatively different at the finest resolution, the overall results on the attractor scale remain consistent (Fig. S6d). Indeed, the attractor decomposition step in STT iteration automatically corrected the clustering bias based on the gene expression similarity or spatial closeness. We have also described the chosen initialization strategy in each dataset in Supplementary Note 2.

Minor Comments:

(1) **Clarity in Method Description:** Some sections could benefit from more detailed explanations, especially concerning the mathematical underpinnings of the transition tensor.

Response: Thank you. In the revised introduction of the main text (line 68-74) and methods description (line 421-426), we have added more explanation of the mathematical concepts involved in STT, including attractor and attractor-specific dynamics in gene regulation models to help readers better understand the background and implication of transition tensor.

(2) **Validation on Diverse Datasets:** A broader range of datasets could provide a more comprehensive performance assessment.

Response: Following your suggestion, we have included the analysis of a new Stereo-Seq dataset on mouse brain (Fig.6) and the pancreas dataset (Fig. S5). Our analysis shows that STT can reveal the spatial attractors in the high-resolution spatial datasets in addition to data generated by the image-based HybISS technology (Fig. 4) and barcoding-based 10X Visium platform with spot resolution (Fig.5).

(3) **Comparison with Existing Methods:** A detailed comparison, especially in terms of accuracy, computational efficiency, and scalability, would be insightful.

Response: Thank you. We have added a detailed comparison of accuracy between STT and scVelo, CellDancer, and UnitVelo (Figure 2b). Our results suggest that the spliced components of STT tensor are in agreement with RNA velocity models, however, the unspliced components which encode the multi-stability feature of attractors exhibit a significant improvement.

We also tested the efficiency and scalability of STT in real datasets. We found that the computation time of each iteration in STT scales linearly both with

respect to number of cells and number of genes (Fig. S9a).

(4)Parameter Sensitivity Analysis: A sensitivity analysis would offer insights into the method's robustness.

Response: Thank you for the suggestion. In the revised Supplementary Figure S6-S8, we have added the sensitivity analysis for mouse brain spatial datasets. The key parameters tested include 1) the weight of tensor-induced kernel 2) the weight of spatial kernel 3) the threshold of multi-stability genes. STT was found to be reasonably robust for its performance within a range of parameters. We observed that

- While different number of attractors K imply various resolution to study the dynamics, the membership-based parameter estimation in STT indeed encourage the consistency between results within certain range of K (Fig.S8)
- The results of STT are robust under different attractor initialization strategy either using clustering of spliced counts, joint spliced/unspliced counts or appropriate cell annotations (Fig.S6).
- While the increasing weight of spatial kernels tend to encourage spatially connected attractors (Fig. S6), the results are robust within a moderate range. In the revised method, we have also incorporated a monitor module to allow users to change the weights of different kernels adaptively.
- The STT attractor decomposition is robust to the change of MS gene threshold in certain range (Fig.S7).

(5)User-Friendly Implementation: The provided GitHub page is a step in the right direction, but there's scope for improvement. Annotated Jupyter notebooks, detailed function descriptions, and interactive tutorials could make the STT method more user-friendly.

Response: Thank you for the suggestion. In the revision, we have reformulated the STT package, for example, incorporating the Scanpy .tl and .pl module style, with detailed function descriptions. We have also developed interactive tutorials with annotated Jupyter notebooks available at <https://github.com/cliffzhou92/STT/tree/release>.

In summation, the manuscript showcases a promising avenue for spatial transcriptomics data analysis. While the method is poised for significant impact, addressing the highlighted concerns can further refine its robustness and applicability.

Response: Thank you again for all your insightful comments and suggestions on our manuscript.

Reviewer #3:

Remarks to the Author:

A. Summary of the key results

This work proposes a computational framework called Spatial Transition Tensor (STT) which extends single cell RNA dynamics into multi-scale, cell-state dependent scenarios, and, with the availability of spatial constraints, it allows for modeling spatial transcriptome datasets as well. Instead of a global equilibrium state assumed by the original single cell RNA velocity method, STT assigns cells into a prespecified number of cell clusters, or attractors, where multiple local, cluster-dependent steady states are assumed to be reached. Such treatment allows the derivation of a gaussian likelihood objective which can be solved analytically. To train STT, two major steps are iterated until cell assignment results converge. Initialized with external cell clusters, in step one STT estimates the rate parameters of the dynamic model from data, before cluster-dependent velocities together with the inter-cellular transition probabilities are estimated. In step two, the dynamic transition matrix based on 1) unspliced and spliced velocities, 2) expression similarities; and 3) an optional spatial kernel for spatial transcriptome is decomposed via spectral clustering method, where the membership of each cell to a cluster is derived. Compared with the original RNA velocity method, there is no need of selecting lower and upper quantile cells for approximating steady-state cells. However, a filtering scheme is needed in each iteration for removing genes that can not well fit the data.

Response: We are very grateful for the careful reading of our manuscript and the insightful comments and suggestions from the reviewer.

B. Originality and significance

Overall, this work is innovative and well-motivated by dynamical systems theory. Conceptually, It contributes to single cell dynamics/trajectory analysis with a more flexible theoretical perspective which is free from the strong assumption made in the original RNA velocity theory; practically, it proposes approximation solutions for model implementation and algorithms for fitting and analysis. Given these contributions, it would be a good resource in the field.

However, although STT is based on dynamical systems theory, the current work did not move further to the validation of its fundamental assumptions on single-cell datasets. As a result, like prior methods, there may still be scenarios where the method falls short, yet we do not have a tool for relevant diagnosis. Nevertheless, the established fundamental conceptual framework as well as the empirical demonstrations are still valuable references which paved the way for future research.

Response: Thank you very much for the positive comments on the originality

of our method. In the revision, we've made major effort in several areas, specifically, we have

- Implemented a monitor module in the algorithm to inspect the iteration process and incorporated the regularization and early-stopping to enhance the robustness of STT;
- Benchmarked the accuracy of STT with four existing methods using quantitative metrics, and highlighted the unique strength of STT to reveal nonlinear attractor dynamics resulting from nonlinear gene regulations;
- Investigated the robustness of algorithm with regard to parameters and initial attractor assignment within certain range, and suggested default parameters;
- Applied STT to high-resolution Stereo-seq mouse brain data;
- Compared the results with CellRank and integrated STT's multi-stability kernel into the CellRank downstream analysis; and
- Improved the illustration and presentation of transition tensors and algorithm overview.

Following the reviewer's suggestions, we have made substantial improvements on the robustness of STT. Please see details below.

F. Suggested improvements

For the manuscript, I have the following comments regarding readability, clarity of plotting, additional benchmark experiments, and hyper-parameter sensitivity analysis.

Introduction

1. Perhaps the underlying dynamics model could be introduced in a more intuitive and self-contained manner, with the assistance of concise cartoons or toy examples grounded in single cell biology, so that non-technical readers can quickly catch the point. The current text may assume prior knowledge of dynamic systems, including concepts like "attractors", "saddle points" as well as the way they are involved in a system.

Response: Thanks for the suggestion. In the revised introduction (line 68-74), we have added more explanation and introduction about dynamical systems concepts and their relevance with gene expression and regulation dynamics to help readers better understand the background of STT.

2. Fig 1.a. and b. seem not very straightforward to understand, without better annotations in the figures or descriptions in the caption/main texts. For example, what do the horizontal lines mean in subfigure STT (Multistable); the visualisation of the transition tensor seems confusing at first glance.

Response: We apologize for the confusion. The horizontal lines indicate the steady state of the unspliced counts in each attractor, and the tensor is composed of gene-specific velocities in each attractor. In the revised Fig.1, we have included explicit annotations both in the figure and the caption to better illustrate the assumption and concept of transition tensor.

Results:

1. General plotting:

The figures need to be improved by providing more helpful visual indicators on the focused patterns, for example, it would be better to have the ground-truth transition directions marked in simulation (Fig. 2) result, and the patterns of interest highlighted in Fig 3-6, especially when cross-figure comparison is needed.

Response: Thanks for the suggestion. In the revised manuscript, we have plotted the ground-truth transition directions in Fig. 2a. In addition, we also added a quantitative comparison with the ground truth velocity by using the cosine similarity metric. We found that while the spliced tensor of STT tensor is in agreement with RNA velocity models, the unspliced tensor which encode the multi-stability feature of attractors, witness a significant improvement.

The arrows look very messy in sub-figures Fig.2b, Fig.4h, k, Fig.6 a-b, Fig.7 f.e, it is sometimes confusing which areas to attend to and compare with.

Response: We apologize for the confusion. In the revised manuscript, we have replotted figures by reducing streamline densities and enlarging figures to better illustrate the arrows.

Cell embeddings are better aligned across different subfigures in comparison, and colored consistently. For example, in Fig 2.a, the colors are not aligned among STT, scVelo, UniT Velo plots, and the Dynamo seems to be plotted in a totally different embedding space. The mismatch makes a lot of cognitive load to compare different methods on the same dataset.

Response: We apologize for the confusion. In the revised manuscript, we have tried to use same embedding when comparing different aspects of the results in Fig.2. For UniT Velo, because it only estimates on the spliced counts, we are using the dimensionality reduction results from spliced counts only. The cells are colored by either 1) STT attractor or 2) clustering results from different methods, thus might have different colors across different figures. We have added the description in corresponding figure caption. Because of the similarity of Dynamo and scVelo on inferring velocity in non-metabolic labelling data, we removed the comparison with Dynamo in the revised manuscript.

2. When doing transition matrix decomposition, what is the motivation of using GPCCA and how is it advantageous over other methods? Given that this is a less popularly used method, it may be helpful to introduce it in more detail.

Response: Thanks for the nice suggestion. GPCCA decomposition can efficiently detect the cell's membership among various attractors, especially when the tensor-induced random walk transition matrix is not in equilibrium – a condition that holds for the majority of cases considering nonlinear gene regulation dynamics. The membership in turn is useful to quantify the uncertainty of each cell's contribution when constructing attractor-wise transition tensors. In the revised manuscript, we have added a more detailed introduction about the mathematical background as well as the biological implication of GPCCA method (line 421-426).

3. There are multiple hyper-parameters involved in STT modeling, but their effects are not reported:

1) A very important hyper-parameter is the number of attractors, or cell clusters, as attractors decide the course-level trajectory as well as local velocities of cells. In empirical analysis, how would the results change with different numbers of clusters, and with different cluster annotations?

2) Weighting of transition matrices for decomposition. How would different weighting schemes affect model learning?

3) Threshold for gene filtering in the iterative model fitting procedure. Empirically, scVelo can produce different results if different sets of velocity genes are selected for cellular transition estimation.

For clarity, there may better be a summary report of all the hyper-parameters applied in different benchmark datasets.

Response: Thank you, we agree that testing the robustness of hyperparameters in STT helps further evaluate the performance of method. In the revised supplementary material, we have tested the influence of 1) number of attractors, 2) attractor initialization strategy, 3) weights in different kernels to construct random walk, and 4) threshold to filter multi-stability genes in each iteration. We observed that

- While different number of attractors K imply various resolution to study the dynamics, the membership-based parameter estimation in STT indeed encourage the consistency between results within certain range of K (Fig.S8)
- The results of STT are robust under different attractor initialization strategy either using clustering of spliced counts, joint spliced/unspliced counts or appropriate cell annotations (Fig.S6).
- While the increasing weight of spatial kernels tend to encourage spatially connected attractors (Fig. S6), the results are robust within a moderate range. In the revised method, we have also incorporated a monitor

module to allow users to change the weights of different kernels adaptively.

- The STT attractor decomposition is robust to the change of MS gene threshold in certain range (Fig.S7).
- We included the summary report of all hyperparameters in Supplementary Table 1 and 2.

4. In line 70- 73, and also line 311-314 [Discussion], there is a claim on the limitation of CellRank and Dynamo based on linear RNA velocity kernels. Concerning this, it can be interesting to also include benchmark datasets on which prior methods were validated, for example, CellRank has included in its package some nice datasets used in their report, e.g., Pancreas and Reprogramming datasets, which are available in the CellRank package <https://cellrank.readthedocs.io/en/latest/api/datasets.html>. It would be nice if STT can show comparable results on these datasets. From another perspective, it might also be interesting to investigate if, based on a multi-stability velocity kernel (e.g. generated by STT), CellRank can improve its performance.

Response: Thanks for the suggestion. In the revised manuscript, we have reanalyzed the pancreas data using STT (Fig. S5). We also modified the STT algorithm such that it will output the multi-stability velocity kernel. We then input the kernel to CellRank for downstream random walk analysis to discover terminal fates and absorption probabilities (Fig. S5b). We found that the fate probability analysis based on STT multi-stability kernel is consistent with CellRank analysis based on RNA velocity to discover terminal fates of alpha, beta and delta cells in pancreas, and they show convergent streamlines of unspliced tensor. In addition, STT analysis also highlights regions in *Fev+* cells as potential attractors. We also simulated the random walks based on multi-stability kernel to highlight the attractors in dataset (Fig. S1d)

Typos:

Line 358: "... and the most likelihood estimation ..." should be maximum likelihood estimation

Line 394: in the equation, the subscript "i" is most likely "k".

Response: Thank you. We have corrected the typos and made a thorough proofreading of the manuscript.

Decision Letter, first revision:

Dear Qing,

Thank you for submitting your revised manuscript "Spatial Transition Tensor of Single Cells" (NMETH-A53055B). It has now been seen by the original referees and their comments are below. The reviewers find that the paper has improved in revision, and therefore we'll be happy in principle to publish it in Nature Methods, pending minor revisions to satisfy the referees' final requests and to comply with our editorial and formatting guidelines.

TRANSPARENT PEER REVIEW

Please note: we allow redactions to authors' rebuttal and reviewer comments in the interest of confidentiality. If you are concerned about the release of confidential data, please let us know specifically what information you would like to have removed. Please note that we cannot incorporate redactions for any other reasons. Reviewer names will be published in the peer review files if the reviewer signed the comments to authors, or if reviewers explicitly agree to release their name. For more information, please refer to our FAQ page.

ORCID

Sincerely,
Madhura

Madhura Mukhopadhyay, PhD
Senior Editor
Nature Methods

Reviewer #1 (Remarks to the Author):

I appreciate the authors' efforts in addressing most of my previous concerns, and I am mostly satisfied with their responses to the major issues raised. The incorporation of a monitoring interface to assess the fitting between spliced and unspliced counts is a commendable improvement. This feature enhances the tool's robustness by allowing users to make necessary adjustments. Additionally, the modifications made to mitigate overfitting are noteworthy. Despite the inherent challenges in visualizing 4D tensors, the improvements in this area, as demonstrated in the manuscript figures, are significant and commendable.

However, I still have reservations regarding the tool's sensitivity to initialization settings. The variations observed in the results under different initial settings, as illustrated in Fig.S6, validate my concerns about the critical impact of initialization on the model. I recommend that the authors provide explicit guidance on selecting the most appropriate initialization settings for various application scenarios. This involves a thorough examination of the tool's robustness across a wide range of scenarios, establishing a general guideline for the best initialization practices. Without such guidance, users may find it challenging to optimize the performance on their datasets, and poor selection of initial settings could significantly impair the model's effectiveness.

Other minor comments:

(1) I suggest adding a section in the documentation that thoroughly describes the API. This section should clearly explain the functions and modules, enabling users to utilize the tool more effectively. For example, a detailed explanation of functions like `st.infer_lineage(adata, si=4, sf=3)`, including their purpose, parameters, and how to set these parameters, would be highly beneficial.

(2) The manuscript would be enhanced by including details about the running time and memory complexity of the tool. It is essential for potential users to understand whether the method is scalable and capable of handling large single-cell datasets. Information about the tool's performance in terms of computational efficiency and resource requirements would assist users in determining its suitability for their specific computational needs and data scales.

I would endorse the publication if the above concerns/comments could be addressed in the revised version.

Reviewer #1 (Remarks on code availability):

The program is generally user-friendly; I was able to install and run it without any issues. However, the documentation could be improved. Specifically, it lacks a detailed API description, which makes it challenging to comprehend the parameters associated with the various modules. This shortfall might hinder the tool's effective application across diverse scenarios.

Reviewer #3 (Remarks to the Author):

My comments are well addressed in this revision.

Reviewer #3 (Remarks on code availability):

I cloned the project STT from the repository, installed the required packages in my server.

I ran two example jupyter notebooks:

- 1) example_toggle.ipynb
- 2) example_emt_circuit.ipynb

The results are generally consistent with authors' outputs, except for some minor differences in estimated parameter values, perhaps, caused by random effect.

Author Rebuttal, first revision:

Reviewer #1 (Remarks to the Author):

I appreciate the authors' efforts in addressing most of my previous concerns, and I am mostly satisfied with their responses to the major issues raised. The incorporation of a monitoring interface to assess the fitting between spliced and unspliced counts is a commendable improvement. This feature enhances the tool's robustness by allowing users to make necessary adjustments. Additionally, the modifications made to mitigate overfitting are noteworthy. Despite the inherent challenges in visualizing 4D tensors, the improvements in this area, as demonstrated in the manuscript figures, are significant and commendable.

However, I still have reservations regarding the tool's sensitivity to initialization settings. The variations observed in the results under different initial settings, as illustrated in Fig.S6, validate my concerns about the critical impact of initialization on the model. I recommend that the authors provide explicit guidance on selecting the most appropriate initialization settings for various application scenarios. This involves a thorough examination of the tool's robustness across a wide range of scenarios, establishing a general guideline for the best initialization practices. Without such guidance, users may find it challenging to optimize the performance on their datasets, and poor selection of initial settings could significantly impair the model's effectiveness.

Response: We again appreciate the reviewer's careful reading and helpful suggestions on our revised manuscript. In the revision, we provided more quantifications and more specific user guide on the initiation choices.

Specifically, in the revised manuscript, we *quantified* the differences among the streamlines using four different initializations for two different datasets (Fig.S7 and S10). We calculated the cosine similarities among averaged tensors obtained by four different initialization strategies: a) region annotation, b) clustering on spliced counts or c) joint counts of both spliced and unspliced, and d) spatial clustering on coordinates. We found that the tensors are generally consistent among different initialization strategies despite minor differences in the attractor details (Fig. S7).

Fig S7b. Sensitivity analysis STT initialization strategy tested on mouse brain HybISS dataset. Under each strategy, for each cell we calculated the cosine similarity between its attractor-averaged tensor (unspliced or spliced component) with the case applied in the main text where the region annotation was used as default. Here "Region" employs the original annotation in dataset about spatial regions, "spa_leiden"

employs the leiden clustering of the spatial coordinates, “exp_leiden” employs the leiden clustering of the gene expression counts, and “joint_leiden” employs the leiden clustering of both spliced and unspliced counts.

Fig S10c. Sensitivity analysis STT initialization strategy tested on chicken heart 10X Visium dataset. Under each strategy, for each cell we calculated the cosine similarity between its attractor-averaged tensor (unspliced or spliced component) with the case applied in the main text where the region annotation was used as default.

We further investigated the sensitivity of hyperparameter “resolution” in Leiden clustering for STT initialization, and we found that the STT tensor outputs are robust to the resolution in the range of the parameter [0.2, 0.6].

Fig S7c. Parameter sensitivity analysis of Leiden clustering resolution for STT initialization tested on mouse brain HybISS dataset. Under each resolution, for each cell we calculated the cosine similarity between its attractor-averaged tensor (unspliced or spliced component) with the case applied in main text where resolution was assigned as 0.3.

To provide more detailed guidelines for the users, in the revised Method section we added a short subsection on the choice of initialization (line 681-691). Specifically, we suggest the users use prior biological annotation of the dataset as an initial assignment for the input of STT. When such information is unavailable, we recommend using the joint clustering outputs implanted in STT based on both spliced and unspliced counts. If the users prefer other cluster methods or more systematic analysis, STT provides an option for direct inputs from the user-generated clustering outputs as the input for the initializations of STT.

In addition, as seen in the updated (expanded) Fig. S6d, when the weight of the spatial kernel is large (≥ 0.5), the found attractors are relatively sensitive to the weight. However, when the weight parameter is chosen around 0.3 (Table S2), STT outputs become more robust. In the revision, we added a systematical sensitivity analysis of spatial weights from 0.1 to 0.4 with a step size 0.05, showing the transition tensor directions are quantitatively consistent in this parameter range. The default value for the weight of spatial kernel is then set as 0.3. In the revised user guide, we explicitly stated that this weight value needs to be chosen between 0.1 and 0.4 (Fig. S6d).

Fig S6d. Parameter sensitivity analysis of spatial weight kernel tested on mouse brain HyBISS dataset. Under each spatial weight, for each cell we calculated the cosine similarity between its attractor-averaged tensor (unspliced or spliced component) with the default case when spatial weight was assigned as 0.3.

Other minor comments:

(1) I suggest adding a section in the documentation that thoroughly describes the API. This section should clearly explain the functions and modules, enabling users to utilize the tool more effectively. For example, a detailed explanation of functions like `st.infer_lineage(adata, si=4, sf=3)`, including their purpose, parameters, and how to set these parameters, would be highly beneficial.

Response: Thank you for the helpful suggestion. In the updated STT package, we have built the documentation webpage at <https://stt-doc.readthedocs.io/en/latest/> including a detailed description of all major functions in API.

(2) The manuscript would be enhanced by including details about the running time and memory complexity of the tool. It is essential for potential users to understand whether the method is scalable and capable of handling large single-cell datasets. Information about the tool's performance in terms of computational efficiency and resource requirements would assist users in determining its suitability for their specific computational needs and data scales.

Response: Thank you for the nice suggestion. In the newly added Supplementary Material Table S3, we have included the running time and memory usage for various datasets analyzed in the manuscript. We found that STT is efficient to execute on a

personal laptop in terms of both computing time and memory usage. As seen in Fig. S10, STT scales linearly with the number of genes and cells when using the subsampling test.

Table S3 Summary table of STT computing time and memory usage for the real datasets on a personal laptop. The peak memory indicates the maximum amount of total memory used during running, within which the incremental memory denotes the memory requested when executing the dynamical analysis API of STT. The memory usage was recorded by the Python memory profiler package. The timing was recorded by taking the average of seven executions by the Python timeit package. The analysis was carried out on a MacBook Air 2023 with M2 chip and 16GB memory.

Datasets	Number of Cells	Number of Genes (input to algorithm)	Running Time (mean +/- std)	Peak Memory/Incremental
A549 EMT	3132	2000	125s± 13.5 s	2039.59 MB/1412.30 MB
Bone marrow	5780	2000	272s± 76 s	3004.10 MB/ 2759.29 MB
Pancreas	2531	2000	100s± 9.44 s	2088.36 MB/ 1573.52 MB
Mouse brain (HybISS)	4628	117	24 s ± 4.97s	847.57 MB/ 502.52 MB
Chicken heart	1967	2000	65s± 8.63 s	1563.92 MB/630.31 MB
Mouse brain (Stereo-seq)	7765	2000	387s± 52 s	5330.27 MB/4350.57 MB

I would endorse the publication if the above concerns/comments could be addressed in the revised version.

Reviewer #1 (Remarks on code availability):

The program is generally user-friendly; I was able to install and run it without any issues. However, the documentation could be improved. Specifically, it lacks a detailed API description, which makes it challenging to comprehend the parameters associated with the various modules. This shortfall might hinder the tool's effective application across diverse scenarios.

Response: Thank you again for all your insightful comments and suggestions on our manuscript. Following your suggestions, in the revision we have improved the manuscript on initialization robustness as well as the detailed documentation on major API of STT.

Reviewer #3 (Remarks to the Author):

My comments are well addressed in this revision.

Reviewer #3 (Remarks on code availability):

I cloned the project STT from the repository, installed the required packages in my server.

I ran two example jupyter notebooks:

- 1) example_toggle.ipynb
- 2) example_empt_circuit.ipynb

The results are generally consistent with authors' outputs, except for some minor differences in estimated parameter values, perhaps, caused by random effect.

Response: We again appreciate the reviewer's careful reading of our revised manuscript and are grateful for the positive comments from the reviewer. In this round of revision, we have further improved the package of STT on initialization robustness as well as the detailed documentation webpage on major API. In addition to the GitHub page, we have also deposited all the processed data and analyzing results to the public link at <https://disk.pku.edu.cn/link/AAC2FC70BC8EEE40ACB6EA524852FFCF23>.

Final Decision Letter:

Dear Qing,

I am pleased to inform you that your Article, "Spatial Transition Tensor of Single Cells", has now been accepted for publication in Nature Methods. The received and accepted dates will be 24th Jul 2023 and 02 Apr 2024. This note is intended to let you know what to expect from us over the next month or so, and to let you know where to address any further questions.

Over the next few weeks, your paper will be copyedited to ensure that it conforms to Nature Methods style. Once your paper is typeset, you will receive an email with a link to choose the appropriate publishing options for your paper and our Author Services team will be in touch regarding any additional information that may be required. It is extremely important that you let us know now whether you will be difficult to contact over the next month. If this is the case, we ask that you send us the contact information (email, phone and fax) of someone who will be able to check the proofs and deal with any last-minute problems.

Please note that *Nature Methods* is a Transformative Journal (TJ). Authors may publish their research with us through the traditional subscription access route or make their paper immediately open access through payment of an article-processing charge (APC). Authors will not be required to make a final decision about access to their article until it has been accepted. Find out more about Transformative Journals

You may wish to make your media relations office aware of your accepted publication, in case they consider it appropriate to organize some internal or external publicity. Once your paper has been scheduled you will receive an email confirming the publication details. This is normally 3-4 working days in advance of publication. If you need additional notice of the date and time of publication, please let the production team know when you receive the proof of your article to ensure there is

sufficient time to coordinate. Further information on our embargo policies can be found here:
<https://www.nature.com/authors/policies/embargo.html>

If you are active on Twitter/X, please e-mail me your and your coauthors' handles so that we may tag you when the paper is published.

Best regards,
Madhura

Madhura Mukhopadhyay, PhD
Senior Editor
Nature Methods